



# Cushion bogs are stronger carbon dioxide net sinks than moss-dominated bogs as revealed by eddy covariance measurements on Tierra del Fuego, Argentina

David Holl[1], Verónica Pancotto[2,3], Adrian Heger[1], Sergio Jose Camargo[3,4], and Lars Kutzbach[1]

[1]Institute of Soil Science, Center for Earth System Research and Sustainability (CEN), Universität Hamburg, Hamburg, Germany
[2]Centro Austral de Investigaciones Científicas (CADIC-CONICET), Ushuaia, Argentina
[3]Universidad de Tierra del Fuego (ICPA-UNTDF), Ushuaia, Argentina
[4] Dirección de Cambio Climático (DCC), Secretaría de Estado de Ambiente, Desarrollo Sostenible y Cambio Climático (SADSyCC), Ushuaia, Argentina

**Correspondence:** David Holl (david.holl@uni-hamburg.de)

**Abstract.** The near-pristine bog ecosystems of Tierra del Fuego in southernmost Patagonia have so far not been studied in terms of their current carbon dioxide ($CO_2$) sink strength. $CO_2$ flux data from southern hemisphere peatlands is scarce in general. In this study, we present $CO_2$ net ecosystem exchange (NEE) fluxes from two Fuegian bog ecosystems with contrasting vegetation communities. One site is located in a glaciogenic valley and developed as a peat moss-dominated raised bog, the other site is a vascular plant-dominated cushion bog located at the coast of the Beagle Channel. We measured NEE fluxes with two identical eddy covariance (EC) setups at both sites for more than two years. With the EC method, we were able to observe NEE fluxes on ecosystem level and at high temporal resolution. Using a mechanistic modeling approach, we estimated daily NEE models to gap-fill and partition the half-hourly net $CO_2$ fluxes into components related to photosynthetic uptake (gross primary production, GPP) and to total ecosystem respiration (TER). We found a larger relative variability of annual NEE sums between both years at the moss-dominated site. A warm and dry first year led to comparably high TER sums. Photosynthesis was also promoted by warmer conditions but less strong than TER with respect to absolute and relative GPP changes. The annual NEE-C uptake was more than three times smaller in the warm year. Close to the sea at the cushion bog site, the mean temperature difference between both observed years was less pronounced, and TER stayed on similar levels. A higher amount of available radiation in the second observed year led to an increase of GPP (5 %) and NEE (35 %) carbon (C) uptake. The average annual NEE-C uptake of the cushion bog (-122 ± 76 g m$^{-2}$ a$^{-1}$, n = 2) was more than four times larger than the average uptake of the moss-dominated bog (-27 ± 28 g m$^{-2}$ a$^{-1}$, n = 2).





## 1   Introduction

Although peatlands cover a comparably small area of the Earth's land surface, they store large amounts of carbon (Yu et al., 2010). Intact peatlands act as net sinks for atmospheric carbon (C) and play an important role in the terrestrial C cycle and therefore in the climate system. While in-depth studies of long-term and recent carbon accumulation rates are available for

many northern hemisphere bogs, carbon flux data from southern hemisphere bogs are scarce. The Magellanic Moorland which covers an area of 44,000 km$^2$ of coastal Patagonia in Chile and Argentina is one of the most notable peatland complexes south of the equator and belongs to the World's largest wetlands (Fraser and Keddy, 2005). Significant parts of the Magellanic Moorland are dominated by a unique type of bog ecosystem which is exclusive to the southern hemisphere. These so called cushion bogs are rain water-fed peatland ecosystems which are dominated by the vascular plants *Astelia pumila* (J. R. Forst.)

Gaudich. (Govaerts, 2019) and *Donatia fascicularis* (J. R. Forst. & G. Forst.) (Ulloa Ulloa et al., 2017) as opposed to the majority of global bogs which are commonly peat moss-dominated. Both cushion plants are characterized by high nutrient use efficiency, slow biomass turnover and a large root biomass in relation to the small aboveground part of the plants (Kleinebecker et al., 2008). Fritz et al. (2011) estimated fine root biomass accumulation to be four times larger in cushion bogs than in northern hemisphere raised bogs. The aerenchymatic roots of vascular cushion plants lead to additional oxygen transport into

the rhizosphere and thereby to highly decomposed peat also near the surface and to close-to-zero methane emissions (Fritz et al., 2011; Münchberger et al., 2019).

In this study, we present carbon dioxide ($CO_2$) flux time series from two bogs located in southernmost Patagonia on Tierra del Fuego. One site is a peat moss-dominated raised bog, the other site is a vascular plant-dominated cushion bog. We measured $CO_2$ fluxes continuously for more than two years at both sites using the eddy covariance (EC) technique. With the EC method,

we were able to measure $CO_2$ net ecosystem exchange (NEE) on ecosystem scale and at high temporal resolution. To date, plant community-scale gas flux measurements in Fuegian peatlands have been conducted within three methane ($CH_4$) flux studies with manual chambers (Fritz et al., 2011; Lehmann et al., 2016; Münchberger et al., 2019). The measurements in these studies were taken at the same sites where data presented in this study were acquired. $CO_2$ flux measurements of any Fuegian ecosystem have not been reported so far. Only recently, multi-annual $CO_2$ (Goodrich et al., 2015a; Campbell et al., 2014) and

$CH_4$ (Goodrich et al., 2015b) flux records have been reported from New Zealand. The ombrothrophic raised bog investigated in these studies is, however, dominated by the rush *Empodisma robustum*, distinguishing it from most other global bogs, which are commonly dominated by peat mosses, and also from the peatlands of Tierra del Fuego.

The overriding research question of this study is: Do the contrasting traits of cushion plants and peat mosses lead to distinct dynamics of primary production and ecosystem respiration both with respect to average annual net C uptake and in relation to

the sensitivity of both bog ecosystems to the variability in environmental driver courses? On that account, our initial objective is to comprehensively describe the $CO_2$ net ecosystem exchange (NEE) flux dynamics of two contrasting bog ecosystems on Tierra del Fuego using an EC setup. We apply a mechanistic modeling approach yielding a partitioned net $CO_2$ flux with two model terms estimating total ecosystem respiration (TER) and gross primary production (GPP) separately. We finally compare




our data with literature records of northern and southern hemisphere NEE balances and whole ecosystem C balances taking into account methane flux data that has been published for the two sites of this study.

## 2 Tierra del Fuego: A review on geography, peatland zonation and research history

### 2.1 Geographical setting and climatic gradients

Tierra del Fuego is an archipelago located at the southern tip of South America between 52° S and 56° S. It is confined by the Magellan Strait in the north and the Beagle Channel in the south (see Figure 1). To a large extend, the landscape and vegetation history of Tierra del Fuego during the Holocene is known from bog profiles, which were analyzed for palaeogeological and paleoecological studies. They show that the transition from the Pleistocene into the Holocene began as early as 17,800 years BP (Rabassa et al., 2000) in the region with glaciers retreating westwards from the terminal moraine location at Punto Moat

(55.0 ° S, 66.8 ° W). Around 11,000 years BP, flooding of the Beagle Channel by the sea began (Vanneste et al., 2015). Open *Nothofagus* woodlands and grasslands developed in the lowlands during the early Holocene when precipitation increased. Variability in rainfall was, however, high, enabling the spread of frequent peat and forest fires during drought periods (Markgraf and Huber, 2010). Precipitation variability and thereby the frequency of fire events decreased after around 5,000 years BP. Along with a change in species composition, dense *Nothofagus* forest replaced the former open woodlands (Markgraf and

Huber, 2010). With shorter summer drought periods, the expansion of ombrothrophic *Sphagnum* bogs was promoted. A climatic shift towards colder, wetter and stormier conditions around 2,600 years BP (Heusser, 1995) led to the invasion of *Sphagnum*-dominated bogs by vascular plants at wind-exposed areas along the north coast of the Beagle Channel on Peninsula Mitre (see Figure 1). These present day cushion bogs are dominated by *A. pumila* and *D. fascicularis*.

  Today, geographical shifts in general peatland types follow the steep climatic east-west gradient across Tierra del Fuego

which is caused by the mountain ranges of the Andes and the Cordillera Darwin and foremost affects the distribution of precipitation. While winds from the west-northwest prevail year-round on the regional scale, the relief divides southern Patagonia in a very moist western part (up to 5000 mm a$^{-1}$) and a steppe (below 300 mm a$^{-1}$) (Tuhkanen (1992) as reviewed in Holl (2017)). However, local precipitation and wind conditions can be heavily influenced by smaller scale relief features of the landscape. At the pacific coast on Islas de los Evangelistas (52°24' S, 75°06 W) for example, the average annual wind speed

is 12 m s$^{-1}$ coming from the northwest, whereas Ushuaia experiences mainly southwesterly winds with an annual average of around 4 m s$^{-1}$ (Tuhkanen, 1992). In general, winds are stronger in spring and summer than in winter. Plant-ecology is, however, not mainly impacted by the speed of the winds but by the constancy with which they sweep across the region. The fact that the strongest winds occur in summer distinguishes southern Patagonia from the mid-latitudes of the northern hemisphere (Weischet, 1985) and puts particular pressure on plants as their trade-off between transpiration and photosynthesis potentially

becomes less beneficial in terms of net C gain during the windy vegetation period. Setting the Magellanic Moorland further apart from nothern hemisphere peatlands and northern ecosystems in general is the low input of airborne anthropogenous pollutants and nutrients. With the Westerlies blowing across southern Patagonia from the open Pacific year-round and due to little agriculture in the region, significant sources of for example heavy metals or nitrogen compounds are comparably far away.




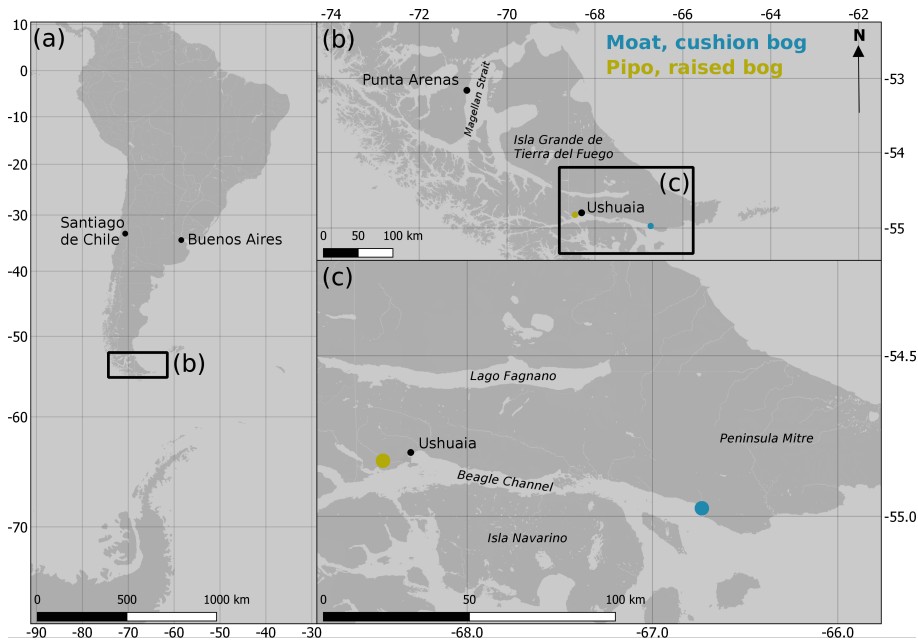

**Figure 1.** Location of the two measurement sites Moat and Pipo along the northern coast of the Beagle Channel on Tierra del Fuego, Argentina. (Map data: OpenStreetMap contributors, under Open Database License, 2019)

Some authors have stressed the importance of Fuegian ecosystems as a basal reference to their anthropocenically altered global counterparts. Studying these landscapes allows for a "glimpse of pre-industrial environments" as Kleinebecker et al. (2008) put it in their biogeochemical analysis of peat samples from an west-east Andean transect (53°S). Fritz et al. (2011) estimated nitrogen deposition at the north coast of the Beagle Channel to be very low at 0.1 g m$^{-2}$ a$^{-1}$ inferring this number from data

published by Godoy et al. (2003) about the Pacific coast of Chilean Patagonia.

In this study, we investigated two ombrotrophic bog sites on Isla Grande de Tierra del Fuego (see Figure 1), both close to the north coast of the Beagle Channel and situated on Argentinean territory. Despite profound similarities between the two ecosystems (both are rain water-fed, peat-accumulating mires), they are dominated by contrasting vegetation communities and occur in different geomorphological settings. According to the vegetation zonation of Moore (1983), one site belongs to the

cushion bogs of the Magellanic Moorland zone, the other one to the *Sphagnum* bogs of the deciduous forest zone.

### 2.2 Fuegian peatland types

The Fuegian landscape as a whole has been termed Magellanic Moorland or Magellanic Tundra Complex (Pisano (1977, 1983) as reviewed in Holl (2017)) and in particular consists of extended wetland areas interlinked with forests. The Magellanic Moorland covers an area of 44,000 km$^2$ (Arroyo et al., 2005) of which 2,700 km$^2$ are located on Argentinean territory (Iturraspe,

2012). In the dry northern and central Isla Grande de Tierra del Fuego, seasonally flooded vegas that lack the presence of peat mosses can be found. Raised *Sphagnum magellanicum* bogs are distributed throughout the central and marginal cordilleran val-



leys and roughly follow the distribution of *Nothofagus pumilio* forests (Tuhkanen (1990) as reviewed in Holl (2017)). Cushion bogs dominate the wet pacific coast but also extensive parts of Peninsula Mitre in the east of Isla Grande and the archipelagic region south of the Beagle Channel on Chilean territory on Isla Navarino.

Cushion bogs form a unique type of peatland, exclusively found on the southern hemisphere. They grow in similar relief
settings like for example Atlantic blanket bogs in Ireland. Their main peat forming species, however, are not mosses but the vascular plants *D. fascicularis* and *A. pumila*. Phylogenetically, *A. pumila* belongs to the Family of *Asteliaceae* in the Order of *Asparagales*. The genus *Astelia* was established by Joseph Banks and Daniel Solander in 1810. *A. pumila* is a perennial herb that grows in dense patches and is characterized by a high ratio of below to above ground biomass (Fritz et al., 2011). Its porous roots are commonly longer than 1 m (Grootjans et al., 2010). With stems of up to 5 cm, the thick and imbricate
1 to 3 cm long and around 0.5 cm wide leafs grow relatively close to the ground. They appear dark green and shiny on the upper, paler and duller on the lower side. The leafs are rigid and show apical growth. Despite its occurrence in nutrient-poor environments, high nutrient use efficiency and low biomass turnover enable *A. pumila* to sustain its dense and relatively large root system (Kleinebecker et al., 2008). Due to additional oxygen being transported into the soil through aerenchymatic roots, the accumulated peat is highly decomposed. Fritz et al. (2011) report humification grades of H8 to H10 on the Von-Post scale
(von Post, 1922) and estimated that cushion bog plants accumulate up to four times more fine root biomass compared to rates from northern hemisphere ombrotrophic bog ecosystem reported by Moore et al. (2002).

The raised *Sphagnum* bogs of the cordilleran valleys on Tierra del Fuego are more similar to northern hemisphere bogs. Biodiversity in these systems is, however, very low in comparison to their northern counterparts. They can be inhabited by as little as 10 vascular plant species (Moen et al., 2005). Their vegetation community consist nearly completely of one moss
species: *Sphagnum magellanicum*. Apart from that, only two other peat mosses commonly occur in transitions to fens (*S. fibriatum*) or in pools (*S. cuspidatum*) as stated by Kleinebecker et al. (2007).

### 2.3 Peatland research on Tierra del Fuego

Tierra del Fuego has been subject to ecological research for centuries. Before Charles Darwin reached the Fuegian archipelago in 1832 aboard the Beagle, Philibert Commerson, Joseph Banks and Daniel Solander who explored Tierra del Fuego during
James Cook's first voyage to the region in 1767 as well as Johann Reinhold Forster and his son Johann Georg Adam Forster who accompanied Cook during his second expedition between 1772 and 1774, have contributed a significant amount of information primarily on the prevalent flora of Tierra del Fuego (Tuhkanen, 1990). Recent descriptions of Fuegian flora are given by Moore (1983) and Roig (1998). A major advance in the scientific exploration of Tierra del Fuego is marked by the expeditions of Väinö Auer on behalf of the Geographical Society of Finnland between 1928 and 1952. Auer conducted a total of 15 extensive
sampling and mapping campaigns in the region. The results obtained with these data cover multiple scientific fields like botany, mesozoic tectonics, quaternary geology, paleoecology, paleoclimatology and volcanology. Auer established the view on Fuegian peatlands how it is predominant in scientific literature until today. He analyzed 70 bog profiles throughout the archipelago focusing on the archive function of peatlands. From the pollen records of the profiles, he drew conclusions about past shifts in ecological boundaries that he attributed to fluctuating climatic conditions, sea level changes and volcanic eruptions. For this





purpose he also used volcanic ash layers preserved in peat and by that pioneered the application of tephrachronology for qua-ternary geology. Since Auer's times, peatlands have often been studied by geologists to yield information on the Pleistocene and Holocene that they contain (e. g. Heusser, 1995; Rabassa et al., 1989, 2000, 2006). More recently, the focus of publications in this discipline has shifted towards the east (Penisula Mitre and Isla de los Estados), beyond the extend of the last glaciation

(Heusser, 1995; Björck et al., 2012; Ponce and Fernández, 2014; Ponce et al., 2016). Data on the occuring peatland types and their distribution across Tierra del Fuego has been gathered and compiled for example by Roivainen (1954), Auer (1965), Pisano (1983), Tuhkanen (1990), Rabassa et al. (1996), Roig et al. (2001), Blanco and de la Balze (2004), Moen et al. (2005), Mauquoy and Bennett (2006), Iturraspe (2010) and Grootjans et al. (2010). Lately, the interest of ecologists in the scarcely studied peatlands of Peninsula Mitre has grown (Fritz et al., 2011; Iturraspe, 2012; Grootjans et al., 2014; Münchberger et al.,

2019) due to their pristine character and their spatial dominance in this part of Isla Grande. Trace gas exchange fluxes of southern Patagonian bogs have been investigated within three $CH_4$ flux studies using data acquired with manual chamber mea-surements from the same sites which we investigated in the study at hand. Fritz et al. (2011) measured $CH_4$ fluxes six times on three days representative for spring, summer and autumn in a cushion bog on Peninsula Mitre close to the Beagle Channel from where a larger summer data set of $CH_4$ emissions has been presented recently by Münchberger et al. (2019). Lehmann

et al. (2016) investigated $CH_4$ fluxes during four summer days from the Río Pipo raised *S. magellanicum* bog near Ushuaia.

## 3 Methods

### 3.1 Site descriptions

#### 3.1.1 Moss-dominated raised bog, Pipo

Río Pipo mire (from hereon Pipo) is located close to the city of Ushuaia at 54.83° S, 68.45° W in Parque Nacional Tierra

del Fuego, 60 m above sea level and around 5 km north of the Beagle Channel. The mire is a raised *S. magellanicum* bog as they are typical for the wind-protected western valleys of Tierra del Fuego (Iturraspe, 2012). It covers an area of around 60 ha at the southern end of a glaciogenic valley bottom. The valley stretches to the SE and is drained by the river Río Pipo, which marks the northern margin of the bog. Along its southern border, a rather narrow lagg zone forms the transition to the adjacent upwards sloping *Nothofagus pumilo* forest. *S. magellanicum* is with a surface cover of around 40 % (Mark et al.,

1995; Lehmann et al., 2016) the most abundant plant species. It occurs in wet lawns and forms roughly N-S oriented chains of hummocks perpendicular to the drainage direction. Alternating strips of lawns with pools and hummocks compose most of the peatland's surface. The drier hummocks are commonly covered by the dwarf-shrub *Empetrum rubrum* and the rush *Marsipospermum grandiflorum* (Lehmann et al., 2016). We found the peat base between 3.8 m and 4.3 m below the surface within four peat cores that we drilled at different microtopographical positions.

The literature account of mean annual precipitation for Ushuaia ranges from 530 mm (Iturraspe, 2012) over 545 mm (Pisano, 1977) to 574 mm (Tuhkanen, 1992). At a central point in the peatland, we measured 515 mm cumulative precipitation between 1 March 2016 and 28 February 2017, between 3 % and 11 % below the literature averages. Due to energy supply outages,





our precipitation records for the remainder of 2017 and 2018 are incomplete. Precipitation sums measured by the Argentinean Servicio Meteorológico Nacional (SMA) at Ushuaia airport (2016: 334 mm; 2017: 418 mm), which is located around four kilometers outside of the city center of Ushuaia on a peninsula in the Beagle Channel, and close to Centro Austral de Investigaciones Científicas (CADIC) in the city of Ushuaia (2016: 462 mm; 2017: 520 mm; personal communication by Gastón

Kreps) also suggest that 2016 was a comparably dry year. Deviations between those sums on the one hand demonstrate the high variability of local precipitation in the region. On the other hand, sensor bias related to the windy conditions at the exposed airport site or to the lack of an accurate representation of precipitation as snow could have contributed to this deviation.

The mean annual temperature at a long-term meteorological station in Ushuaia is 5.5 °C (Iturraspe, 2012). In 2017, we measured an annual mean air temperature of 5.3 °C. Mean June air temperatures were 5.8 °C in 2016 and 0.0 °C in 2017.

Compared to the long-term average June temperature of 1.2 °C given by Iturraspe (2012), the winter of 2016 was particularly warm whereas June of 2017 was closer to the mean. Maximum annual air temperatures occured on 26 March in 2016 (21.7 °C) and on 6 November in 2017 (22.1 °C), minimum air temperatures on 18 August in 2016 (-5.6 °C) and on 17 June in 2017 (-8.1 °C). During both years, wind came almost exclusively from west-northwestern directions, hence from the valley.

### 3.1.2 Vascular plant-dominated cushion bog, Moat

The second site (from hereon Moat) is located close to Bahía Moat at 54.97 °S and 66.73 °W. The bay is formed by the creek Río Moat that drains towards the south into the Beagle Channel at the south-western edge of Penisula Mitre. Our measurements were conducted in a cushion bog approximately 1 km off the Beagle Channel's northern coast which developed on a series of three glaciofluvial plains elevated between 33.1 m and 40.3 m above mean sea level (Borromei et al., 2014). The bog is limited by Río Moat in the west and a Pleistocene fronto-lateral moraine in the south (Borromei et al., 2014). It is sickle-shaped,

covers an area of 170 ha and slopes at 0.6° slightly towards the south-east. In the north, the mire is bordered by Subantarctic Evergreen Forest dominated by *Nothofagus betuloides* and *Drimys winteri* (Heusser, 1995) interlinked with *S.magellanicum* peatlands. The cushion bog drains into a channel in the north that partly runs below-ground and receives water from both the bog and a hill in the north. The outflow channel joins with several other creeks further east before discharging into the Beagle Channel around 4 km south-east of Bahía Moat.

More than 70 % (Fritz et al., 2011) of the bog's surface is covered by the evergreen cushion plants *A. pumila* and *D. fasciculares* that form dense and firm lawns. They occur together with *Caltha dioneifolia, C. appendiculata, Carex antarctica, Drosera uniflora, Empetrum rubrum, Tetroncium magellanicum* and stunted *Nothofagus spp.*. Although the mircrorelief of this landscape is not very pronounced, a pattern of cushions and small ponds has developed (Blanco and de la Balze, 2004). Small areas, often on the somewhat wind-protected edges of pools, are dominated by *S. magellanicum*. In the peat profile, however,

remnants of *S. magellanicum* make up large parts of the material under areas that cushion plants dominate today. We found *Sphagnum* peat from the peat base at 7 m until 4.2 m below the surface. From 4.2 m to 2.2 m, thin layers of alternating decomposition grade and detritus amount that lack cushion plant remnants occur. From 2.2 m to the surface, the substrate consists of highly decomposed *A. pumila* peat. At the same site, other studies report similar results. Heusser (1995) found *A. pumila* residuals until 1.7 m below the surface, Fritz et al. (2011) report *Sphagnum* peat at depths greater than 3 m. Borromei





et al. (2014) give a peat depth of 9.96 m that the authors date at $9.750 \pm 40$ years BP. Long-term weather data is not available for Punto Moat. With our setup, we measured 576 mm cumulative rainfall between 1 February and 31 December 2016 and an annual precipitation of 726 mm in 2017. Mean annual air temperature was 6.27 °C in 2016 (February to December) and 6.38 °C in 2017. Mean June air temperature was considerably higher in 2016 (5.41 °C) than in 2017 (2.07 °C). With a mean

February value of 8.90 °C in 2016 and 10.35 °C in 2017 summer air temperatures were more similar between the years. The highest air temperatures were measured on 26 March (22.44 °C) in 2016 and on 6 November (22.48 °C) in 2017, coldest air temperatures on 2 July (-4.18 °C) in 2016 and on 17 June (-7.53 °C) in 2017.

## 3.2  Instrumentation

Each eddy covariance (EC) system we used to estimate turbulent $CO_2$ fluxes consisted of a 3D sonic anemometer (Windmas-

ter Pro, Gill, UK), a infra-red gas analyzer and a data logger (LI-7200 and LI-7550; Licor, USA). Additional atmospheric variables were recorded on a separate data logger (CR-3000; Campbell Scientific, UK). Air relative humidity and temperature were measured with a HC2-S3 probe (Rotronic, CH), photosynthetically active radiation $PAR$ with a SKP-215 sensor (Skye Instruments, UK), precipitation with a ARG 100 raingauge (EML, UK). The instrumental setup was identical at both sites. Gas concentrations and three-dimensional wind velocity raw data were logged at 20 Hz between 31 January 2016 and 17 May 2018

in Moat and between 08 February 2016 and 17 April 2018 in Pipo. Biomet data were recorded at 1 Hz within the same time spans. Both LI-7200 analyzers were running on factory calibration until 11 November 2017 (Moat) and 09 November 2017 (Pipo), were zero- and span-calibrated and restarted on 17 November 2017 (Moat) and 16 November 2017 (Pipo). Energy to run the equipment was generated on-site with a wind turbine (LE 600; Leading Edge, UK; peak power 600 W) and two photovoltaic panels (OS-172; Leading Edge, UK; peak power 85 W).

## 3.3  Data processing

### 3.3.1  Flux calculation and quality-filtering

We used the EC technique to determine half-hourly gas and energy fluxes from the high-frequency raw gas concentration and three-dimensional wind velocity time series. A comprehensive description of the EC approach is given for example by Aubinet et al. (2012). Half-hourly turbulent $CO_2$ fluxes were computed using the software EddyPro 6.2.0 (Licor, USA) and included

the following (standardized, see Holl (2017); Holl et al. (2019)) steps.

We detected and removed raw data spikes according to Vickers and Mahrt (1997), with a maximum of 1 % accepted spikes and a maximum of three samples as consecutive outliers. Because we used anemometers that were affected by a firmware bug (published as Technical Key Note KN1509v3 by Gill Instruments), we compensated for the apparent underestimation of the vertical wind speed measurement by instructing EddyPro to apply the multiplication factor given in the mentioned publication.

Moreover, due to the use of Gill anemometers, we applied an angle of attack correction, i.e. a compensation for flow distortion induced by the anemometer frame (Nakai et al., 2006). Coordinate rotation to align the anemometer x-axis to the current mean streamlines was calculated as double rotation according to Kaimal and Finnigan (1994), linear detrending as proposed by Gash



and Culf (1996). With simultaneously available water vapor concentration, cell temperature and cell pressure measurements from the LI-7200 gas analyzer, $CO_2$ concentrations could be converted directly into mixing ratios, i.e. concentrations referring to dry air of constant temperature (Ibrom et al., 2007b; Burba et al., 2012), making corrections for air density fluctuations unnecessary.

To determine time lags between the water vapor concentration measurements and the vertical wind speed time series, we used the automatic time lag optimization option in EddyPro. For this procedure, prior to processing the complete dataset, time lags were determined for sub-periods of raw data with varying instrumental setup by covariance maximization (Fan et al., 1990). A searching window around the median of the found time lags (nominal timelag, $T_{nom}$) is defined by $T_{nom} \pm 3.5 \times MAD$, where $MAD$ is the median absolute deviation of the found time lags. When processing the complete dataset, EddyPro performed a

covariance maximization of vertical wind speed and the scalar of interest for each half hour and then checked whether the found time lag fell within the searching window defined before. If not, $T_{nom}$ was used as time lag. Water vapor concentration time series were binned in ten relative humidity classes. The procedure was applied to each class, resulting in ten different nominal time lags. Time lags between the $CO_2$ concentration and vertical wind speed time series were estimated by covariance maximization. We calculated time lag statistics and a nominal time lag using the above described option in EddyPro. $CO_2$ time

lags were not divided in different humidity classes. We addressed time lag statistics later during quality-filtering.

We corrected for low-frequency loss due to finite averaging time and linear detrending as described by Moncrieff et al. (2004). In order to obtain a correction factor for each flux value, EddyPro estimated true cospectra as proposed by Kaimal et al. (1972) and reformulated by Moncrieff et al. (1997) for each half hour. We set EddyPro to remove high-frequency noise from the gas concentration spectra before continuing with the spectral attenuation estimation. In this step, a lower limit of the expected

high-frequency noise was user-set. The software then linearly interpolated between this lower limit and the highest available frequency (20 Hz) in the log-log transformed spectrum and substracted this function from the spectrum's high-frequency part. Lower limits were set to 5 Hz for $CO_2$ and water vapour concentration spectra. After noise-removal, frequency-wise multiplication with a transfer function yielded an estimate of the filtered signal in the frequency domain. The transfer function was selected by EddyPro according to the used detrending method as given by Moncrieff et al. (2004). After integrating over

the averaging period, a low-cut spectral correction factor for each raw flux was calculated.

We accounted for high-frequency loss due to path averaging, signal attenuation and finite time response of the instruments using the method of Fratini et al. (2012). EddyPro first determined the cut-off frequency ($f_c$) and natural frequency ($f_n$) by fitting the amplitude response of a first-order low-pass filter ($H_{IIR}(f_n/f_c)$) to power spectrum ensembles of the respective scalar time series. For water vapour, $f_c$ was estimated for nine relative humidity classes. Correction factors $F1$ were calculated

with two methods depending on sensible and latent heat fluxes being above (high fluxes) or below (low fluxes) the thresholds of 10 and 5 $\mathrm{Wm}^{-2}$ respectively (Fratini et al., 2012). For high fluxes, EddyPro calculated the correction factors as proposed by Hollinger et al. (1999). $F1$ estimation included the degradation of the unattenuated sensible heat flux cospectrum by multiplying it with $HIIR(f_n/f_c)$ for the previously determined $f_c$. For low fluxes, the obtained $F1$, $f_c$ dataset was fitted to the model given by Ibrom et al. (2007a) for stable and unstable conditions. We performed cut-off frequency and function param-

eter estimation on ensembles of (co)spectra seperately for two subperiods that we divided at the calibration date (see section



'Instrumentation'). Before using them for ensemble spectra estimations, the single (co)spectra were quality-filtered using the scheme of Vickers and Mahrt (1997) and by omitting half-hours that were assigned quality class 2 according to Mauder and Foken (2004). We corrected for spectral losses due to crosswind and vertical separation between the LI-7200 tube intake and the anemometer in EddyPro following Horst and Lenschow (2009). Additionally, we set EddyPro to calculate random flux

uncertainty estimates (Finkelstein and Sims, 2001) and three quality flags as proposed by Mauder and Foken (2004) which represent flux quality in values from 0 to 2 with 0 denoting the highest quality class. This quality evaluation is based on tests for stationarity and developed turbulence and thereby indicates whether general EC assumptions about atmospheric conditions were met during a flux calculation period.

We performed the following sequence of quality-filtering steps on the $CO_2$ fluxes: We evaluated sensor diagnostics by using

the relative signal strength indication (RSSI) logged from the gas analyzer. Fluxes associated with RSSI values below 65 were discarded. Furthermore, we excluded fluxes when an half-hourly time lag, determined with covariance maximization, fell outside the time window around $T_{nom}$ that was defined by running the automatic time lag optimization routine in EddyPro (see section 'Flux calculation'). $T_{nom}$ as well as the window size were determined for subperiods of different analyzer calibration state. To remove remaining outliers, we defined a range of accepted fluxes by using the 0.1st and 99.9th percentile of the

Mauder and Foken (2004) quality class 0 and 1 fluxes as thresholds.

### 3.3.2   Net ecosystem exchange partitioning model

Unintentional as well as maintenance-related system outages and quality-filtering of the calculated $CO_2$ fluxes led to gaps in the net ecosystem exchange (NEE) data sets. The time series from Moat spans 40176 half hours with about 12 % missing records. The Pipo time series is comprised of 38353 thirty minute steps containing 46 % NEE gaps while this higher percentage of gaps

is mainly related to more frequent power outages due to insufficient input through the wind turbine and the photovoltaic panels. To be able to calculate annual C balances from the measured NEE data sets, we gap-filled the half-hourly time series with a mechanistic modeling approach. We fitted the function given by Runkle et al. (2013) to our NEE, air temperature and radiation data. This bulk model approach is based on the combination of a hyperbolic light saturation function (Thornley, 1998; Zheng et al., 2012) to represent photosynthesis and an exponential temperature–respiration relation (Van't Hoff, 1898). At times of

missing NEE observations, the estimated model parameters can be used together with temperature and $PAR$ data to estimate NEE fluxes. Moreover, the net flux can be partitioned in its components total ecosystem respiration (TER) and gross primary production (GPP).

$$NEE(T, PAR) = R_{base} \times Q_{10}^{\frac{T-T_{ref}}{\gamma}} - \frac{P_{max} \times \alpha \times PAR}{P_{max} + \alpha \times PAR} = TER(T) + GPP(PAR) \tag{1}$$

We optimized the four parameters maximum photosynthetic rate $P_{max}$, base respiration $R_{base}$ (both in $\mu mol\, m^{-2}\, s^{-1}$), temper-

ature sensitivity coefficient $Q_{10}$ and initial quantum yield $\alpha$ (both dimensionless) using a non-linear least absolute residuals method in Matlab (version 9.5). This algorithm is comparable to a traditional non-linear least squares method, the cost function that is minimized during optimization is, however, not the sum of squared but of absolute residuals, reducing the effect of outliers. We used air temperature $T$ (°C) and photosynthetically active radiation $PAR$ ($\mu mol\, m^{-2}\, s^{-1}$) as independent variables





and the quality-filtered $CO_2$ fluxes of Mauder and Foken (2004) classes 0 and 1 as dependent variable NEE. We set the reference temperature $T_{ref}$, at which $TER(T_{ref}) = R_{base}$, to 15 °C and $\gamma$ to 10 °C following Runkle et al. (2013) and Mahecha et al. (2010). We divided the measured time series in one, two and five day intervals and estimated a set of four parameters for each time window. We used bounds ($P_{max}$: [0 30]; $\alpha$: [0 0.05]; $R_{base}$: [0 5]; $Q_{10}$: [1 3]) and start points ($P_{max}$: 5; $\alpha$: 0.02; $R_{base}$:

1.5; $Q_{10}$: 1.4) to constrain parameter optimization. Parameter uncertainty was estimated using their 95 % confidence bounds.

In addition to the estimation of NEE fluxes at times when filtered observed fluxes were not available, the bulk modeling approach allows for a decomposition of the net $CO_2$ flux in contributions related to photosynthesis and respiration. Before using the bulk model parameter estimates (see Eq. (1)) for the calculation of modeled NEE, GPP and TER time series, we quality-filtered the parameter time series and applied smoothing functions to them. We rejected all parameters of a time window

when any of the parameter uncertainty estimates was larger than the respective parameter value itself or the algorithm could not ascertain an error estimate indicating a parameter value being stuck at its upper or lower bound during iterative optimization. To smooth the parameter time series we used a locally weighted regression (Lowess/Loess) method (Cleveland, 1979) in Matlab. Lowess fitting yields new estimates for the input data that are closer to being members of a continuous function than the input series. Lowess smoothing is a stepwise process during which a linear function is fitted to subsets of data around one focal point

for which a new value is estimated as the output of the fitted function. Line fitting comprises the assignment of weights to all used points that increase with their distance from the focal point along the x-axis (time). The use of second order polynoms for locally weighted regressions is commonly denoted by the slightly differing abbreviation Loess. We used Loess smoothing and quality-filtered results from bulk model estimates of two day windows for all parameter time series except for $Q_{10}$ for which we used five day window bulk models and first order polynoms during smoothing. The number of points included in each polynom

fit during Lowess or Loess is referred to as span and was always set to 30 % of all available points. As quality-filtering of the bulk model parameter estimates resulted in unequally spaced parameter time series, we interpolated the smoothed parameter values linearly to one day intervals before driving the bulk models.

We assume that seasonal changes in plant-physiological characteristics are more likely to follow a continuous function than to exhibit rapid variations. Our method of smoothing the bulk model parameter time series is effective in the removal of noise

introduced by bulk model fitting that leads to scattering in the parameter time series. Noise characteristics do change with the window size used during bulk model fitting, larger windows lead to less scatter. Window size is then acting as a low-pass filter that potentially smears up possibly inherent parameter changes on shorter time scales that are, in contrast, still intact after Lowess/Loess smoothing. Shorter term variations, for example synoptic-scale changes of environmental conditions, can lead to short term adaption of plants that should be reflected in bulk model parameter time series. The changes in Moat $P_{max}$ curvature

in summer 2016/2017 (see Figure 2) are an example of the smoothed parameter series' ability to reflect inter-annual variations of the seasonal parameter course, in this case a prolonged phase of unusually cloudy conditions in Moat during early summer in December (see Appendix, Figure C1).



To estimate a $PAR$ value at which the canopy photosynthetic potential was reduced to one tenth of its initial value $\alpha$, and therefore near saturation, we calculated $PAR_{\text{sat}}$ as

$$PAR_{sat}(P_{max}, alpha) = \frac{P_{max}}{\alpha} + \frac{\sqrt{4\frac{P_{max}^2}{\alpha^2} - 4P_{max}^2\left(\frac{1-z_{sat}}{\alpha^2}\right)}}{2} \qquad (2)$$

using the previously determined $P_{\text{max}}$ and $\alpha$ time series at one-day intervals and an attenuation factor $z_{\text{sat}}$ of 10. To set up Eq.
(2), we differentiated Eq. (1) with respect to $PAR$ and set it equal to a by one tenth attenuated quantum use efficiency. We solved this equation for $PAR$ to calculate $PAR_{\text{sat}}$ time series for both sites. Details are given in appendix D.

### 3.3.3 Processing of ancillary meteorological variables

Before half-hourly NEE simulation, we filled gaps in the thirty minute $PAR$ and air temperature records. We applied a mean diurnal variation method similar to the approach of Falge et al. (2001) which exploits the commonly high auto-correlation of
these meteorological variables. Missing values were replaced by averages of available records at the same hour of day within increasing time windows between one and seven days around a gap. Window size was increased until at least one record could be found. If more than one record was available, an average was used to fill the respective gap. Whithin the Pipo data set, 7 % of meteorological observations were missing, and 92 % of these values were filled with averages. The Moat time series contained 1 % gaps, in 93 % of all cases more than one record was found within the time window around a gap. Gap filling
was mostly (70 %) performed using one day windows for the Pipo data set and with one to three day windows in most cases (57 %) of the Moat time series.

Data analysis included the calculation of the cumulative temperature and radiation quantities growing degree days ($GDD$) in °C and cumulative photosynthetically active radiation $PAR_{\text{cum}}$ in $mol\ m^{-2}$. We defined $GDD_5$ as the sum of all positive differences between daily average temperatures and a reference temperature which we set to 5 °C. $GDD_5$ was calculated
as monthly sums for comparison with monthly respiration sums. We expressed $PAR_{\text{cum}}$ as daily sums for comparison with cumulated daily GPP amounts.

### 3.3.4 Calculation of annual net ecosystem exchange sums

We proceeded to drive daily versions of Eq. (1), each comprised of a distinct set of four parameters, with half-hourly, gap-filled radiation and temperature data. We calculated uncertainty estimates $u_{GPP}$, $u_{TER}$ and $u_{NEE}$ for each modeled GPP, TER and
NEE flux based on gaussian error propagation and representing a 95 % confidence interval. We took the partial derivatives of Eq. (1) with respect to the four parameters as described in the appendix (Eq. (A1) to Eq. (A3)). We simplified the process by neglecting comparably small random errors in temperature and radiation measurements. We used the root mean squared error (RMSE) of the smoothed daily parameter time series and the quality-filtered bulk model parameter assessments as uncertainty estimates for each value of the smoothed parameter series.
The partitioned NEE time series from Moat contains data from 29 months (838 days), the Pipo data set includes records from 27 months (800 days). For the analysis of annually accumulated fluxes, we only used months for which gap-filled data




was available at each half hour of each day. We used these 27 (Moat) and 25 (Pipo) complete months to investigate inter-annual variability at each site as well as general variability between the sites.

We constructed an average annual course from spring (1 September) to winter (31 August) by calculating mean monthly sums of NEE, GPP and TER (see Figures B1, B2 and B3). Complete monthly records were mostly available from two years,

three full monthly records were available for March at Pipo as well as for February, March and April at Moat. We expressed the range of individual monthly sums as standard deviation. In most cases, when n = 2, this means that the range is expressed as the absolute difference between both values divided by the square root of two. To estimate and compare average annual courses of the vertical $CO_2$ balances between both sites, we cumulated the twelve average monthly sums. We also added together the standard deviations related to each of these sums to indicate the impact of variations between years on the annual balances. By

adding the monthly range estimates and not using the root of the sum of their squares we treated them as potentially systematic rather than random variations of annual NEE, GPP and TER sums.

To investigate the effect of changing environmental conditions on the partitioned NEE balances at both sites, we divided the time series in two consecutive years that we will term Y1 and Y2 from hereon. Because the observations spanned a bit more than two years at both locations, we additionally checked the effect of choosing different start dates for twelve month intervals

that represent various versions of Y1 and Y2. We defined three start dates (Pipo: 01 March, 01 April; Moat: 01 Febuary, 01 March, 01 April), yielding two (Pipo) and three (Moat), in large parts overlapping versions of Y1 and Y2 per site.

Estimated GPP and TER fluxes contain only bulk model results, NEE fluxes include measurement data of quality class 0 and modeled fluxes at times without highest quality observations. Random uncertainty of the observed fluxes was estimated using the method of Finkelstein and Sims (2001) (see flux calculation section), modeled fluxes were assigned the previously

determined $u_{GPP}$, $u_{TER}$ and $u_{NEE}$ values as uncertainty estimates. The uncertainty of the annual balances was calculated by taking the square root of the sum of squared individual thirty minute flux uncertainties.

## 4 Results and discussion

### 4.1 Quality-filtering

Quality-filtering of the measured data resulted in 5 % and 13 % omitted records from Moat and Pipo respectively. Most fluxes

(1786 and 3105) were filtered out due to the time lag detection quality filter. Of the remaining 36363 and 22037 points, 81 % and 71 % were of Mauder and Foken (2004) quality class 0; 97 % and 95 % of combined quality classes 0 and 1.

Quality-filtering of the bulk model parameter time series resulted in 290 $P_{\max}$, $\alpha$ and $R_{\mathrm{base}}$ values from two-day-window fits for Moat and 155 for Pipo. Related to the total number of two-day windows spanned by the time series, data coverage amounts to 69 % and 34 % for Moat and Pipo respectively. In case of the five-day intervals used for the estimation of $Q_{10}$ time series,

85 and 140 values met the quality criteria equaling 84 % and 53 % of all five day intervals within the NEE time series.





## 4.2 Bulk model parameter time series

We applied Lowess/Loess smoothing to the parameter time series and thereby generated new sets of bulk model parameters. Smoothed and original bulk model parameters are compared in Figure 2. Results show a generally good agreement with the highest coefficient of determination ($r^2$) of 0.8 for $P_{\max}$ at both sites, $r^2$ values between 0.5 and 0.7 for $\alpha$ and $R_{\text{base}}$ and lowest
$r^2$ values of 0.2 in case of $Q_{10}$. Loess/Lowess model uncertainty, estimated as RMSE, ranges between 22 % and 29 % of the respective mean parameter value across both sites and parameter sets. Bias errors, expressed as sum of differences between bulk model and smoothed data points divided by the number of samples, are negative and small for both sites (Pipo: $P_{\max}$: -0.03 µmol m$^{-2}$ s$^{-1}$, $\alpha$: -0.00008, $R_{\text{base}}$: -0.01 µmol m$^{-2}$ s$^{-1}$, $Q_{10}$: -0.002; Moat: $P_{\max}$: -0.03 µmol m$^{-2}$ s$^{-1}$, $\alpha$: -0.00006, $R_{\text{base}}$: -0.007 µmol m$^{-2}$ s$^{-1}$, $Q_{10}$: -0.003)
General differences in ecosystem characteristics between the cushion plant-dominated and moss-dominated site are indicated by the lower level of $P_{\max}$, $R_{\text{base}}$ and $\alpha$ values throughout the courses of both years in Pipo. Moreover, the timing of the slope changes within these parameters series demonstrate that cushion plant photosynthetic activity reaches its maximum earlier in summer compared to the moss-dominated community, while inter-annual variations also lead to variations of this time lag. While $P_{\max}$ maxima were reached about six weeks earlier in Moat (28 November 2016) than in Pipo (13 January 2017) in
summer of Y1, Moat $P_{\max}$ maxima (8 January 2018) were about three weeks ahead of Pipo (28 January 2018) in summer of Y2. $R_{\text{base}}$ maxima time lags were less pronounced and amounted to about ten days in Y1. In contrast to the timing of $P_{\max}$ maxima, $R_{\text{base}}$ maxima were reached earlier at Pipo. In summer of Y2, Moat and Pipo base respiration developed virtually concurrently. While $Q_{10}$ Lowess estimates are the least certain compared to the three other parameters, smoothing reveals a contrasting, mirrored behavior of the sensitivity of respiration to temperature changes between both sites. While $Q_{10}$ maxima
occur in winter and minima in summer throughout all years in Moat, Pipo $Q_{10}$ reaches its maxima in summer and its minima in winter. However, the $Q_{10}$ value ranges we found are small, which is in line with results from Mahecha et al. (2010) who reported that ecosystem level temperature sensitivity of TER is with $1.4 \pm 0.1$ rather stable across different global ecosystems.

## 4.3 Flux gap-filling and partitioning

After Lowess/Loess smoothing, we interpolated the bulk model parameter time series to equally spaced one day intervals and
proceeded to drive those daily models with half-hourly meteorological data. We used the resulting NEE time series to gap-fill the observed data sets enabling us to estimate annual vertical partitioned $CO_2$ balances of both bogs. The gap-filled NEE data sets consist of quality class 0 measured fluxes and 26 % (Moat) and 59 % (Pipo) modeled fluxes. Agreement between eddy covariance net $CO_2$ fluxes and modeled fluxes is high while RMSEs are low (Moat: 0.74 µmolm$^{-2}$s$^{-1}$; Pipo: 0.50 µmol m$^{-2}$ s$^{-1}$) as illustrated in Figure 3. The models cannot explain a relatively small amount of measured high positive
fluxes at both sites which could possibly be related to the rapid release of bubbles from ponds (ebullition, see e.g. Glaser et al. (2004)). Deviations between model and measurement data increase with increasing absolute fluxes and are smaller close to zero. Bias errors (BEs), expressed as sum of differences between measured and modeled data points divided by the number of samples, are positive but low for both sites (Moat: 0.05 µmol m$^{-2}$ s$^{-1}$, Pipo: 0.04 µmol m$^{-2}$ s$^{-1}$). The much smaller value





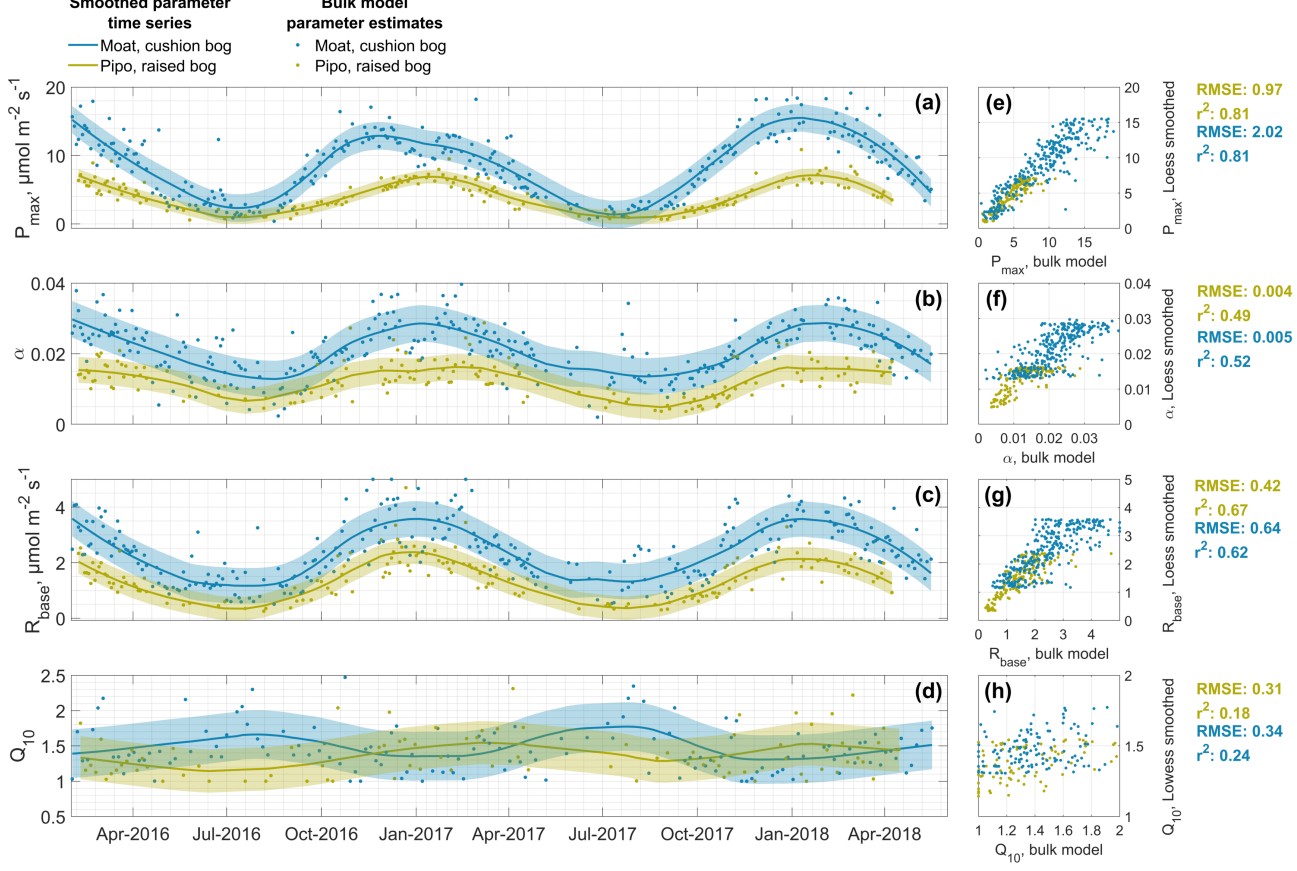

**Figure 2.** Time series of the parameters maximum photosynthesis $P_{max}$ (a), initial quantum yield $\alpha$ (b), base respiration $R_{base}$ (c) and temperature sensitivity $Q_{10}$ (d) that we estimated with two-day, and in case of $Q_{10}$ five-day, window NEE bulk models (dots) and smoothed with a locally weighted regression (Lowess/Loess) method (lines). Areas around lines indicate the uncertainty of smoothed parameter values. Correlations between original bulk model estimates and smoothed values are shown in panels (e) to (h) including the coefficients of determination $r^2$ and root mean squared error RMSE.

range of $CO_2$ fluxes from Pipo compared to Moat is also apparent in the abovementioned figure. Furthermore, we determined correlation coefficients between the modeled TER time series and nighttime (PAR < 10 μmol m$^{-2}$ s$^{-1}$) EC NEE fluxes and found that also this partitioned flux was well explained by our model (Moat: $r^2$ = 0.8, n = 14887, RMSE = 0.43 μmol m$^{-2}$ s$^{-1}$, BE = -0.003 μmol m$^{-2}$ s$^{-1}$; Pipo: $r^2$ = 0.6, n = 6081, RMSE = 0.40 μmol m$^{-2}$ s$^{-1}$, BE = -0.08 μmol m$^{-2}$ s$^{-1}$).

### 4.4 Inter-annual flux and driver variability

To investigate the inter-annual variability of GPP and TER, we cumulated the modeled half-hourly time series annually for different versions of Y1 and Y2 as described above (section 3.3.4). To estimate changes in cumulative NEE fluxes, we used gap-filled EC fluxes and summed them up over the same time periods. We calculated mean annual balances from the different





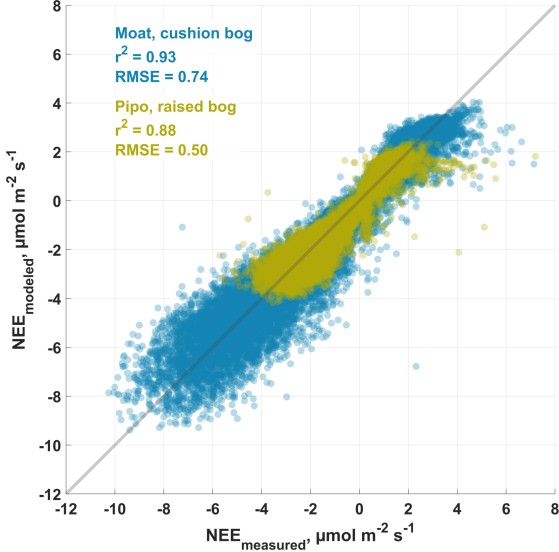

**Figure 3.** Scatter diagram of modeled half-hourly measured eddy covariance net ecosystem exchange (NEE) fluxes and modeled NEE fluxes which were derived using 1. Coefficient of determination ($r^2$) and root mean squared error RMSE are given for both investigation sites (Pipo, n = 15903; Moat, n = 29648).

versions of Y1 and Y2 (see Table 1). To put the balances into context with shifts in meteorological drivers of photosynthesis and respiration, we calculated the monthly cumulatated temperature and radiation measures $GDD_5$ and $PAR_{cum}$ (see Figures C1 and C2 in the appendix).

In 2016, growing season temperatures peaked in March, about one month later than in the two other observed summers.
This vegetation period shift continued with an unusually warm late summer and autumn and culminated in an extremely warm winter month of June at both sites. Mean June temperatures at Pipo were 4.6 °C above the long-term average (Iturraspe, 2012). Only at Pipo, early summer of the next vegetation period (November 2016, still in Y1) was once again warmer than the average. With respect to cumulatively available heat, these temperature variations led to a decrease of $GDD_5$ of around 16 % at Pipo in Y2, whereas $GDD_5$ rose by less than 1 % at Moat which is located closer to the coast. The raised bog's ecosystem respiration
was modulated stronger by the warmer conditions in Y1 than photosynthesis that dropped only by around 2 % in Y2. However, photosynthesis seemingly was promoted by the larger availability of heat in Y1 as well, as despite the cumulative radiation increase by about 4 % in Y2 at Pipo, cumulative |GPP|[1] still dropped. The annual $PAR$ sum increased stronger, by about 10 %, at the cushion bog site where we estimated a 5 % rise in cumulative |GPP|. As mentioned before, close to the coast, the summer of Y1 apparently was considerably more cloudy than the subsequent season (see Figure C1 in the appendix). From Y1
to Y2 , net $CO_2$ uptake increased on average by 35 % in Moat and more than tripled in Pipo. In terms of absolute $CO_2$ uptake increase, the rise at Moat was, however, larger than at Pipo, denoting the fact that $CO_2$ fluxes are on a lower level at Pipo in

---

[1]Following micrometeorological conventions, we use GPP with a negative sign as the associated $CO_2$ flux is directed from the atmosphere towards the surface. We use absolute values to be able to describe a decrease/increase of the process photosynthesis with a decrease/increase of |GPP|





general. Y1 was an extreme year, especially at Pipo, where prolonged warm temperatures (see Figure C1 in the appendix) and reduced precipitation likely led to dry topsoil conditions that promoted heterotrophic respiration diminishing the ecosystem's $CO_2$ sink function in this year.

**Table 1.** Annual net ecosystem exchange (NEE), total ecosystem respiration (TER) and gross primary production (GPP) $CO_2$ fluxes at both investigation sites for different one-year time spans within the observed period between February 2016 and April 2018. Accumulated fluxes and uncertainties are expressed as carbon flux in g m$^{-2}$ a$^{-1}$. Random uncertainties of measured fluxes were estimated using the method of Finkelstein and Sims (2001), modeled flux uncertainties were calculated by propagating individual random uncertainties through Eq. (1) (see appendix A). Moat is the cushion bog site, Pipo the raised bog site in this study. (Jan: January; Feb: February; Mar: March; Apr: April)

|  |  | 2016/2017 (Y1) | | | 2017/2018 (Y2) | | |
|---|---|---|---|---|---|---|---|
|  |  | TER | GPP | NEE | TER | GPP | NEE |
| Moat | 1 Feb – 31 Jan | 645 ± 4 | -757 ± 3 | -105 ± 9 | 644 ± 4 | -764 ± 3 | -103 ± 12 |
|  | 1 Mar – 28 Feb | 649 ± 4 | -746 ± 3 | -90 ± 9 | 651 ± 4 | -786 ± 3 | -115 ± 12 |
|  | 1 Apr – 31 Mar | 649 ± 4 | -738 ± 3 | -82 ± 9 | 661 ± 4 | -810 ± 3 | -129 ± 12 |
| Pipo | 1 Mar – 28 Feb | 392 ± 3 | -399 ± 2 | -9 ± 7 | 339 ± 3 | -385 ± 2 | -32 ± 6 |
|  | 1 Apr – 31 Mar | 389 ± 3 | -397 ± 2 | -8 ± 7 | 343 ± 2 | -390 ± 2 | -35 ± 6 |

The values of all cumulative NEE fluxes and their components that were used to calculate averages from different versions
of Y1 and Y2, distinguished by various start months, are given in Table 1. The largest difference of annual NEE sums between Y1 and Y2 arises in case of the Moat data set when February and March 2016 are excluded from Y1 and accordingly the same months of 2018 are included in Y2. Late summer 2015/2016 was characterized by cool temperatures and less available radiation than summer 2017/2018 when we measured the highest radiation sums and modeled highest cumulative |GPP| whithin the observed time series. Excluding an extreme event in the first year and at the same time including a period with an opposite
effect on the net $CO_2$ flux in the second year maximizes the differences between both years at both sites. The choice of a particular start month used for summing has a larger impact on annual C uptake at Moat than at Pipo. While NEE-C uptake changes from Y1 to Y2 range from a decrease of 2 g m$^{-2}$ to an increase of 47 g m$^{-2}$ at Moat, the choice of different start months leads to less variability (NEE-C uptake increases from Y1 to Y2 between 23 g m$^{-2}$ to 27 g m$^{-2}$) at Pipo.

Apart from differences in inter-annual variations of annual NEE sums between the sites, distinctions also arise with respect
to the variability of monthly sums from different years. Variations within monthly NEE component sums (see Figures B1, B2 and B3 in the appendix) are most pronounced at Moat with respect to GPP and at Pipo with respect to TER. In December, the |NEE|-C differences between 2016 and 2017 are large at both sites (Moat: 32 g m$^{-2}$; Pipo: 15 g m$^{-2}$). Although $GDD_5$ was considerably larger only at Moat in December 2017, the relative rise of TER from December 2016 to December 2017 was small (1 g m$^{-2}$). At Pipo, where $GDD_5$ increase was smaller from December 2016 to December 2017, TER-C loss dropped
by 8 g m$^{-2}$. November of 2016 was particularly warm at Pipo, effects might have carried into December. The monthly $PAR$ sum increased from December 2016 to December 2017 at both sites and led to an increase of GPP-C uptake of 33 g m$^{-2}$ at Moat and of 7 g m$^{-2}$ at Pipo. Monthly NEE-C balances of Pipo appear to deviate more intensively from an average annual



**Table 2.** Average change from the first to the second observed year of cumulated $CO_2$ net ecosystem exchange NEE and its components gross primary production GPP and total ecosystem respiration TER at both sites. Variations of environmental drivers between both years are represented by changes in the temperature sum growing degree days $GDD_5$ and in cumulative photosynthetically active radiation $PAR_{\mathrm{cum}}$. We used averages of the values given in Table 1 in detail to create this simplified overview. (incerease/decrease of less than 5 % (+/–), between 5 and 50 % (++/– –) and more than 50 % (+++/– – –)

| | 2016/2017 (Y1) → 2017/2018 (Y2) | |
| --- | --- | --- |
| | Moat, cushion bog | Pipo, raised bog |
| $PAR_{\mathrm{cum}}$ | ++ | + |
| $GDD_5$ | + | – – |
| \|GPP\| | ++ | – |
| TER | + | – – |
| \|NEE\| | ++ | +++ |

course when deviations from a mean annual temperature course occur concurrently. At Moat, the GPP response to deviations from the mean in the available amount of $PAR$ has an overriding effect on variations of monthly NEE-C balances. While TER is generally at a higher level at Moat compared to Pipo, inter-annual temperature variations within months have a less pronounced relative impact on the mean seasonal course of TER.

5 On average, net $CO_2$ uptake increased substantially at both sites from Y1 to Y2. However, this similar net change resulted from contrasting magnitudes and directions of changes of GPP and TER at both sites. Table 2 gives a simplified, coarsely abstracted overview of the differences in averaged cumulated flux and driver quantities between Y1 and Y2 and from site to site, detailed results are given in Table 1 and Figures C1 and C2 in the appendix. From Y1 to Y2, the magnitude of GPP and TER increased at Moat while |GPP| and TER decreased at Pipo. The reduction of cumulative respiration at Pipo was, however,
10 larger than the drop of the |GPP| sum, leading to an increase in annual net $CO_2$ uptake. At Moat, the rise in |NEE| traces back to an increase in |GPP| that was larger than the simultaneous ascent of respiration.

### 4.5 Average annual fluxes

As a way to ascertain general site differences in $CO_2$ flux patterns, we used all available full months to construct average annual courses of NEE and its components GPP and TER. We calculated monthly sums of all fluxes (see Figures B1, B2 and
15 B3) and averaged these sums across all available records of each month. We summed up these monthly averages over one year from 1 September to 31 August to obtain mean annual courses as described in the methods section. Resulting annual sums, including the sums of the standard deviations of each mean month sum, are shown in Figure 4. On average, the cushion bog takes up 4.5 times more $CO_2$ than the moss-dominated bog, resulting from twice as much photosynthetic uptake and only 80 % higher respiratory $CO_2$ loss. Moat develops its $CO_2$ sink function much earlier in the year in mid-spring (October) whereas
20 net $CO_2$ uptake starts at Pipo only in summer (January). Cumulative C loss through respiration starts to level off in Pipo in late autumn whereas respiration rates stay on more stable levels from mid-autumn throughout winter in Moat. Photosynthesis





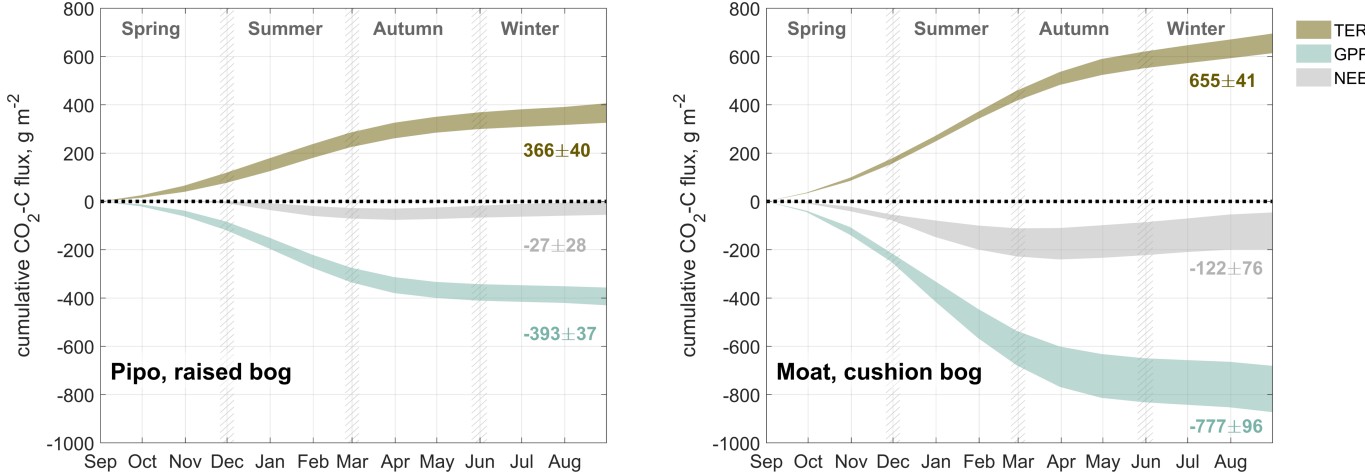

**Figure 4.** Comparison between average annual courses of cumulated $CO_2$ net ecosystem exchange (NEE) flux and its components total ecosystem respiration TER and gross primary production GPP. Ranges denote the accumulated variability of month sums between the different observed years.

continues throughout winter at both sites with lowest rates occurring in June and July. At Moat, GPP starts to reach late autumn (May) rates again in August and the cumulative GPP curve slopes much steeper during spring and summer than at Pipo, where May GPP levels are reached again one month later in September.

## 4.6 Flux–driver relations

To characterize the two bogs' effectiveness in the use of total available $PAR$ to fix $CO_2$, we estimated a cumulative quantum yield $\phi_{cum}$ as the absolute slope of a linear regression between daily $PAR$ and GPP sums. A quantum yield quantity in general is unitless as it describes the amount of fixed moles of $CO_2$ per moles of absorbed photons per area and time. $\phi_{cum}$ describes this photon use efficiency on a daily cumulative basis. In contrast, the initial quantum yield $\alpha$ that is commonly used in plant-physiological studies, and also one of the parameters in our bulk model approach, describes the quantum use efficiency at initial

light levels. Due to the non-linear response of photosynthesis to $PAR$ level, $\alpha$ values are about one order of magnitude larger than $\phi_{cum}$ values. The ratio between the cushion bog's and the raised bog's properties is similar with respect to both light use efficiency measures. The cushion plant-dominated communities in Moat are about twice as efficient in initial (see Figure 2) and cumulative (see Figure 5) photon use compared to the moss-dominated communities in Pipo. Based on the higher photon use efficiencies and the generally higher $P_{\max}$ levels at Moat, we hypothesized that cushion plants are able to still use additional

radiation input effectively when moss photosynthesis is already at light saturation. To substantiate the assumption that cushion plant-dominated communities use additional photon input at high light levels more efficiently, we used Eq. (2) to estimate a $PAR_{\text{sat}}$ time series (see Figure 6) of $PAR$ levels at which photosynthesis was near light saturation (at a tenth of its initial quantum yield $\alpha$). During the vegetation periods, $PAR_{\text{sat}}$ values at Moat mostly exceeded those at Pipo and mostly also the





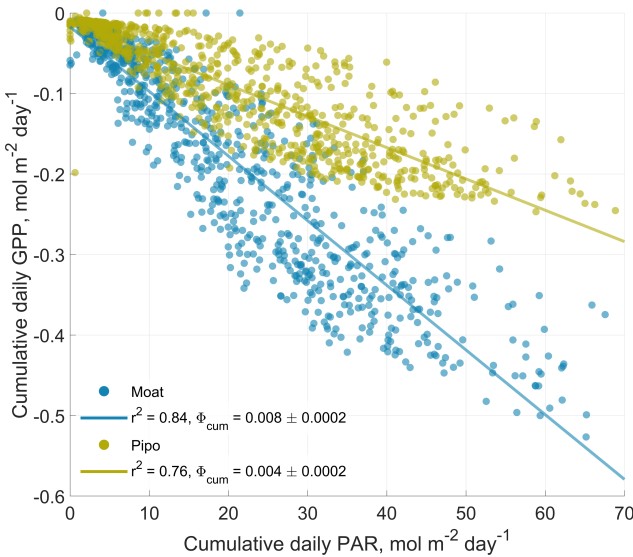

**Figure 5.** Response of daily gross primary production (GPP) sums to the sums of available photosynthetically active radiation ($PAR$) per day. The cumulative quantum yield $\phi_{cum}$ is defined as the absolute slope of a linear regression (ordinary least squares method) between both variables and denotes the average amount of fixed $CO_2$ per amount of absorbed photons per area and time.

amounts of incoming radiation (see Figure 6), confirming our hypothesis. The moss-dominated communities appear to be progressively more adapted to using larger radiation input with $PAR_{sat}$ peaks late in summer. In contrast, the cushion plant-dominated community's ability to efficiently use high $PAR$ input peaks in early summer, even earlier than radiation amounts reach their maximum, and drops off throughout the vegetation period. Winter GPP under lower cumulative light conditions

stays at a higher level at Moat than at Pipo (see Figure B1) due to the higher initial quantum yield of the cushion bog vegetation community (see Figure 2). In contrast, during the normal (with mean temperatures close to the long-term average) winter of 2017, $PAR_{sat}$ at Pipo was higher than at Moat. Additionally, winter $PAR_{sat}$ courses were more similar between 2016 and 2017 at Pipo. The warmer winter of 2016 did affect $PAR_{sat}$ levels at Moat, which exceeded those at Pipo during this cold season. The moss-dominated communities are therefore in principle able to use higher $PAR$ levels in cool periods more effectively

than the cushion plant-dominated communities. Taking into account the low amount of available radiation (see Figure 6), the moss-dominated communities could, however, not profit from this beneficial trait as photosynthesis-saturating-levels of $PAR$ were never reached in any of the two observed winter periods. Photosynthesis limitation by temperature is more pronounced at Moat than at Pipo, but during a period of the year when radiation is the overriding limitation anyhow.

We investigated the effect of prolonged warm periods on ecosystem respiration by inspecting possible functional relation-

15 ships between $GDD_5$ and monthly TER sums. Taking into account the higher number of parameters by calculating the corrected Akaike information criterion $AICc$, second order polynomial fits yielded the best model performances (see Figure 7). These function shapes suggest an $GDD_5$ optimum for TER and a limitation of TER during the warmest months. The parabola vertex and thereby the $GDD_5$ optimum was surpassed during the warmest months at Pipo, whereas it lies just outside the





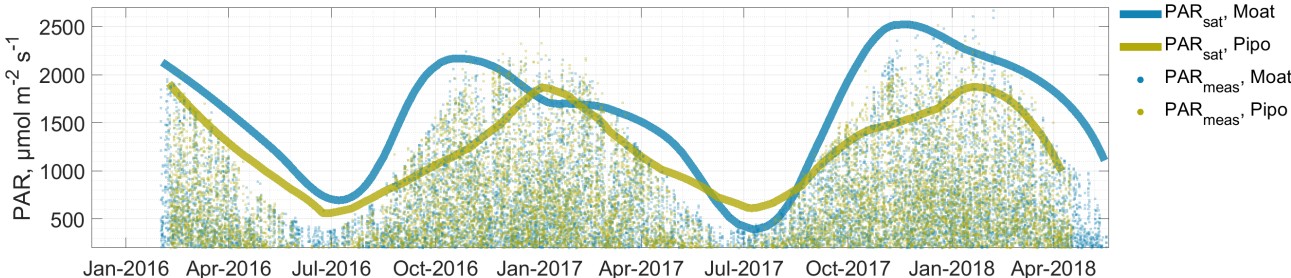

**Figure 6.** Modeled $PAR_{sat}$ time series indicating the value at which photon use efficiency reaches one tenth of the initial quantum use efficiency for the cushion plant-dominated site at Moat and the moss-dominated site at Pipo. Half-hourly measured $PAR$ time series are also shown ($PAR_{meas}$).

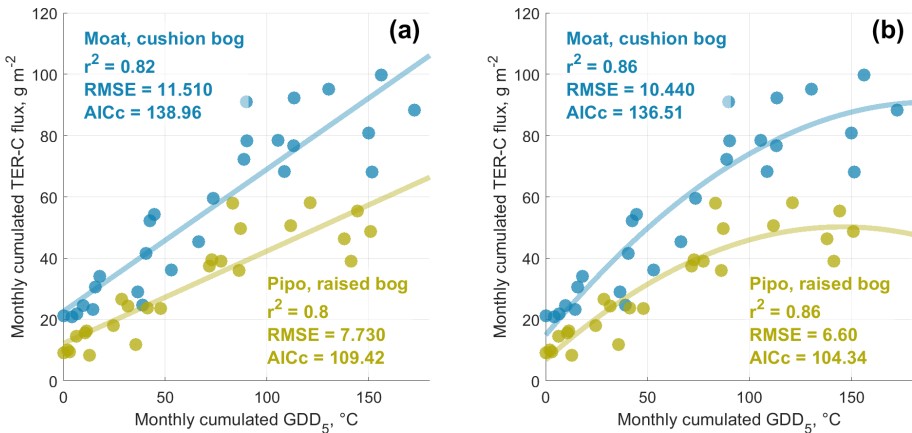

**Figure 7.** A linear (a) and a second order polynomial (b) fit between monthly cumulated growing degree days (GDD) and monthly total ecosystem respiration (TER) sums. While the coefficient of determination ($r^2$) is higher for the parabolic fit, corrected Akaike information criterion (AICc) values and root mean squared errors (RMSEs) are lower than for the linear fit at both sites. Better model performances of second order polynomial fits suggest a $GDD_5$ optimum of TER at both sites.

observed $GDD_5$ range at Moat. Overall, TER in Moat is at a higher level also during cold periods ($GDD_5 = 0$) and increases stronger with additional heat input than in Pipo.

## 4.7 Comparison with other bog ecosystems

We compared the NEE balances of this study to results from similar ecosystems. In case of the Pipo data set, we used literature data from a northern hemisphere ombrotrophic raised bog in Canada. Similarities to other wetland types are less pronounced in case of the Moat data set. We evaluated our results in reference to southern hemisphere vascular plant-dominated bogs in New Zealand as well as to Atlantic blanket bogs in Ireland. To represent $CO_2$ flux dynamics of a typical raised *Sphagnum* bog we inspected results from the moss-dominated ombrotrophic raised bog Mer Bleu in Ontario, Canada from where extensive flux

time series exist. Roulet et al. (2007) estimated cumulative NEE-C fluxes based on multi-annual EC time series from the moss-dominated Mer Bleu bog to average at -40.2 $\pm$ 40.5 g m$^{-2}$ a$^{-1}$. The mean net $CO_2$ uptake of the raised bog at Pipo is within the same order of magnitude but comparably low. Similar to what we found in this study at the *S. magellanicum*-dominated site, inter-annual variability at Mer Bleu is high (-2 to -112 g m$^{-2}$ a$^{-1}$). Large inter-annual variability and even switches from

NEE-C source to sink, as also indicated in our study, are also known from other global peatlands (Arneth et al., 2002; Aurela et al., 2002). NEE components from Pipo are both of lower magnitude than reported by Moore et al. (2002) from Mer Bleu where |GPP| is about 25 % and TER about 30 % larger.

With respect to long-term C sequestration, Loisel et al. (2014) reported average Holocene rates of 22.9 $\pm$ 2.0 g m$^{-2}$ a$^{-1}$ for northern hemisphere ombrotrophic and minerotrophic peatlands. In a meta-analysis of peat core data from mostly peat

moss-dominated bogs in Patagonia, Loisel and Yu (2013) found average Holocene C-accumulation rates of 16 g m$^{-2}$ a$^{-1}$ indicating a reduced C uptake rate of Patagonian bogs compared to average northern hemisphere peatlands. NEE cannot be compared directly to long-term C uptake rates as the C balance is comprised of further flux components like lateral transport of dissolved organic carbon (DOC) or vertical $CH_4$-C flux. Methane flux data is available for both sites in this study from Lehmann et al. (2016) and Münchberger et al. (2019). Taking into account the different surface cover of various vegetation

communities, Lehmann et al. (2016) report a mean daily, area-weighted $CH_4$ flux of 13.1 $\pm$ 7.4 mg m$^{-2}$ d$^{-1}$ referring to the whole bog area on a summer day at Pipo. Cumulating this daily flux over one year yields an evidently overestimated annual $CH_4$-C flux of 3.59 $\pm$ 2.03 g m$^{-2}$ a$^{-1}$ that is, however, still lower than the average $CH_4$-C flux of 3.7 g m$^{-2}$ a$^{-1}$ measured by Roulet et al. (2007) at Mer Bleu. It stands to reason that the lower plant productivity we observed at the Fuegian *Sphagnum* bog is correlated with lower $CH_4$ emissions compared to Mer Bleu. Putting our result of an average annual NEE of -27 $\pm$

28 g m$^{-2}$ a$^{-1}$ into context with the long-term C accumulation of 16 g m$^{-2}$ a$^{-1}$ that Loisel and Yu (2013) determined for Patagonian bogs, also requires an approximation of lateral DOC transport. For northern hemisphere raised bogs, DOC loss has been estimated to range around 10 g m$^{-2}$ a$^{-1}$ (Roulet et al., 2007; Moore et al., 2002). Taking into account this rough approximation of DOC flux and the comparably small $CH_4$-C fluxes at Pipo (Lehmann et al., 2016), the NEE-C balance we determined in this study appears conceivable when compared to long-term Holocene C accumulation rates from Loisel and Yu

(2013).

Annual C flux data from other vascular plant-dominated southern hemisphere bogs has so for been published by Campbell et al. (2014) and Goodrich et al. (2015a, b) from Kopuatai bog on the north island of New Zealand (37°55.5' S) which is dominated by the wire rush *Empodisma robustum* and other species from the family *Restionaceae* with sporadic appearance of *Sphagnum* mosses. Two years of EC $CO_2$ fluxes reported by Campbell et al. (2014) show NEE-C sums between -250.3 and

-218.2 g m$^{-2}$ a$^{-1}$. Net $CO_2$-C uptake of these New Zealand bogs was therefore about twice as large as at the Fuegian cushion bog investigated for the present study while also cumulative |GPP| and TER fluxes are both about 30 % larger. In contrast to the Fuegian cushion bog, seasonal variability of the photosynthetic potential at the *Empodisma robustum*-dominated bog in New Zealand is small (Goodrich et al., 2015a). Methane C loss at the latter site was with up to 21.75 g m$^{-2}$ a$^{-1}$ (Goodrich et al., 2015b) considerably larger than at the studied Fuegian cushion bog where soil–atmosphere $CH_4$ exchange is very low (Fritz

et al., 2011; Münchberger et al., 2019). The intense transport of oxygen through the dense root system of *A. pumila* results in





low CH$_4$ fluxes. Cumulating mean daily summer fluxes measured by Münchberger et al. (2019) at Moat yields an once again most likely overestimated annual CH$_4$-C flux of 0.39 ± 0.70 g m$^{-2}$ a$^{-1}$ for cushions dominated by *A. pumila* and an even lower flux sum of 0.23 ± 0.25 g m$^{-2}$ a$^{-1}$ from the pools between cushions. On the other hand, CH$_4$-C fluxes from the seldom *S. magellanicum* patches at Moat are with 6.66 ± 4.82 g m$^{-2}$ a$^{-1}$ larger than at Pipo.

Atlantic blanket bogs that develop in similar relief settings but receive more than three times more precipitation (Laine et al., 2006) compared to Fuegian cushion bogs show generally lower annual NEE-C uptake rates as reported by Sottocornola and Kiely (2005, 2010) and Koehler et al. (2011) from a blanket bog in Ireland (51 °55' N). Based on six years of EC data, the latter authors estimated mean annual NEE-C fluxes of -47.8 ± 30.0 g m$^{-2}$ a$^{-1}$, about half of the net uptake rate at Moat. Coastal blanket bogs at higher latitudes exhibit a smaller net CO$_2$ sink strength as estimated by Lund et al. (2015) at a boreal blanket

bog in Norway (68 °08' N). There, mean annual NEE-C fluxes based on five years of EC measurements amounted to -19.5 ± 18.3 g m$^{-2}$ a$^{-1}$. With 4.1 ± 0.5 g m$^{-2}$ a$^{-1}$ (Koehler et al., 2011) CH$_4$-C flux, blanket bog CH$_4$ emissions are comparable to what has been determined for northern hemisphere raised bogs and much higher than in Moat (Münchberger et al., 2019).

## 5   Conclusions

The two investigated Fuegian bog ecosystems are located in relatively close proximity to each other but at contrasting topo-

graphic positions. The moss-dominated raised bog developed in a depressed glaciogenic valley bottom, whereas the cushion bog established directly at the coast of the Beagle Channel close to a terminal moraine position of the last ice age and covers more uneven, sloped terrain. The proximity of the cushion bog to the sea especially leads to deviations in the local weather conditions between the cushion bog and the moss-dominated bog in the valley. Contrasting vegetation communities established at the sites: The raised bog is dominated by the peat moss *S. magellanicum* and the cushion bog by the vascular plant *A. pumila*

which is characterized by a dense root system.

Both bogs show distinct surface–atmosphere CO$_2$ flux dynamics. Average annual net CO$_2$-C uptake at the cushion bog is with a factor of 4.5 clearly larger than at the raised *S. magellanicum* bog. Site differences arise with respect to CO$_2$ net exchange and relating to photosynthetic uptake as well as to ecosystem respiration. Both of the latter NEE components are of larger magnitude at the cushion bog: TER is 80 % larger, |GPP| twice as large. Relative inter-annual variability is higher at the

raised bog but substantial at both sites. The raised bog responded to the comparably warm and dry first of the two observed years with a large relative increase of TER and a smaller rise of GPP. Although more cumulative radiation was available in the cooler second year, GPP did not increase at the raised bog. In contrast to the conditions at the raised bog, cumulative temperature expressed as GDD was very similar in both years at the cushion bog. Close to the sea, the summer of the first year was particularly cloudy, and photosynthetic CO$_2$ uptake increased substantially in the second year under conditions of higher

cumulative PAR availability.

In summary, the raised bog's NEE balance is within the typical range of data from similar moss-dominated northern hemisphere raised bogs, while it belongs to the less productive varieties. Compared to global moss-dominated bogs, the investigated Fuegian cushion bog exhibits a much larger net CO$_2$ uptake. Closer to the equator on the southern hemisphere, vascular plant-





dominated bogs exist in New Zealnad that, while emitting more $CH_4$, take up even more net $CO_2$ per year. The cushion bog in this study generally took up about a factor of two more net $CO_2$-C per year than Atlantic blanket bogs that occur in similar geomorphological settings on the northern hemisphere. Although cushion bog ecosystems are exclusive to the southern hemisphere, they are an integral part of one of the largest global wetland complexes (Fraser and Keddy, 2005), the Magellanic

Moorland. It extends outside of Tierra del Fuego to the north along the Pacific coast of Chile and covers an area of 44,000 $km^2$ (Arroyo et al., 2005). Future, regional-scale estimates of land–atmosphere $CO_2$ exchange fluxes from the Magellanic Moorland should take into account the apparently higher primary production and higher ecosystem respiration of cushion bogs compared to raised bogs that both form this large wetland complex. The higher productivity of cushion plants could also be part of an explanation for the apparent competitive advantage cushion plants had over peat mosses when they invaded Fuegian *Sphagnum*

peatlands during a shift of climatic conditions in the region around 2,600 years BP.

*Data availability.*  Data is available for download on the AmeriFlux website (https://ameriflux.lbl.gov/sites/siteinfo/AR-TF1, https://ameriflux.lbl.gov/sites/siteinfo/AR-TF2)





## Appendix A: Bulk model error propagation

$$u_{GPP} = 2 \times \sqrt{\left(\frac{\delta GPP}{\delta P_{max}}\right)^2 \times u_{P_{max}}^2 + \left(\frac{\delta GPP}{\delta \alpha}\right)^2 \times u_\alpha^2} = 2 \times \sqrt{\left(\frac{\alpha^2 PAR^2}{(P_{max} + \alpha PAR)^2}\right)^2 \times u_{P_{max}}^2 + \left(\frac{P_{max}^2 PAR}{(P_{max} + \alpha PAR)^2}\right)^2 \times u_\alpha^2}$$

(A1)

$$u_{TER} = 2 \times \sqrt{\left(\frac{\delta TER}{\delta R_{base}}\right)^2 \times u_{R_{base}}^2 + \left(\frac{\delta TER}{\delta Q_{10}}\right)^2 \times u_{Q_{10}}^2} = 2 \times \sqrt{\left(Q_{10}^{\frac{T-15}{10}}\right)^2 \times u_{R_{base}}^2 + \left(R_{base} \times \frac{T-15}{10} \times Q_{10}^{\frac{T-15}{10}-1}\right)^2 \times u_{Q_{10}}^2}$$

(A2)

$$u_{NEE} = 2 \times \sqrt{u_{GPP}^2 + u_{TER}^2}$$

(A3)



## Appendix B:  Cumulative bulk model results

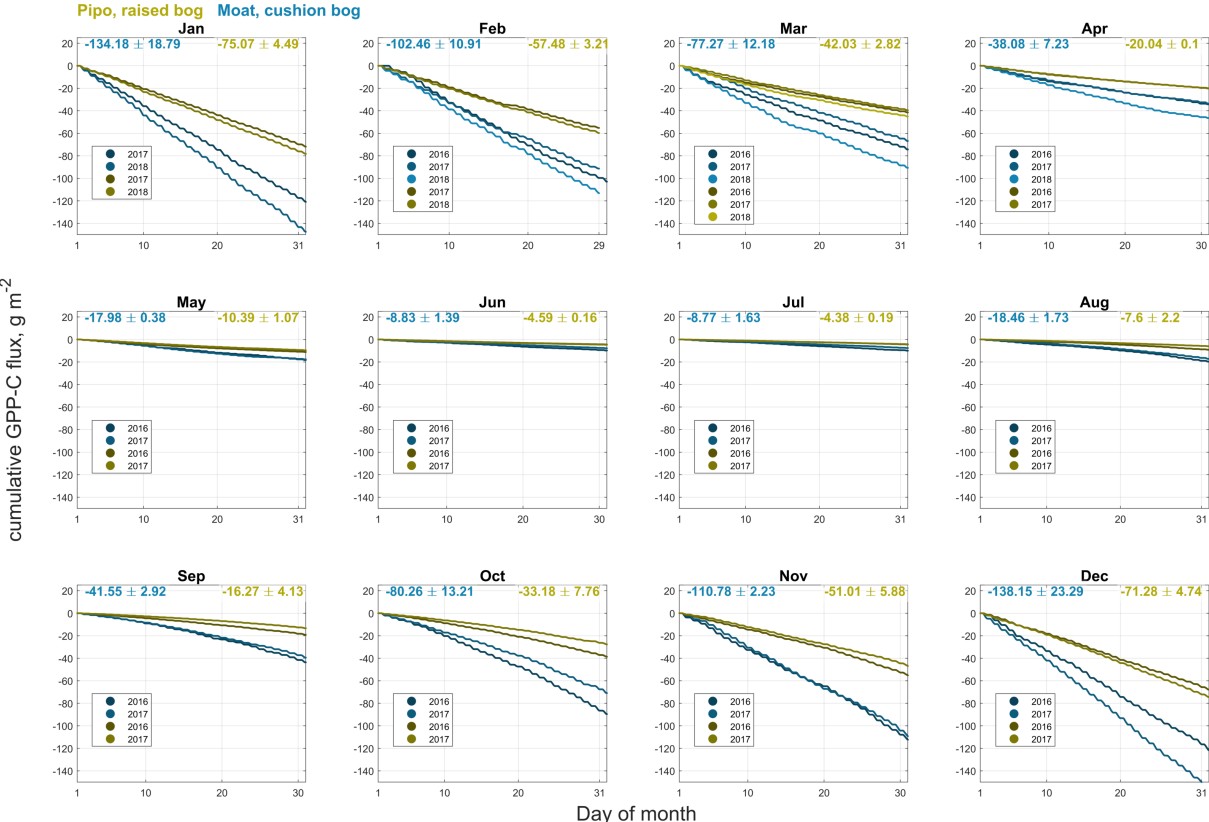

**Figure B1.** Cumulative gross primary production (GPP) CO$_2$ fluxes including standard deviations for monthly intervals of all available full months within the years 2016, 2017 and 2018 at both investigation sites (yellow: Pipo, moss-dominated bog; blue: Moat, cushion plant-dominated bog).



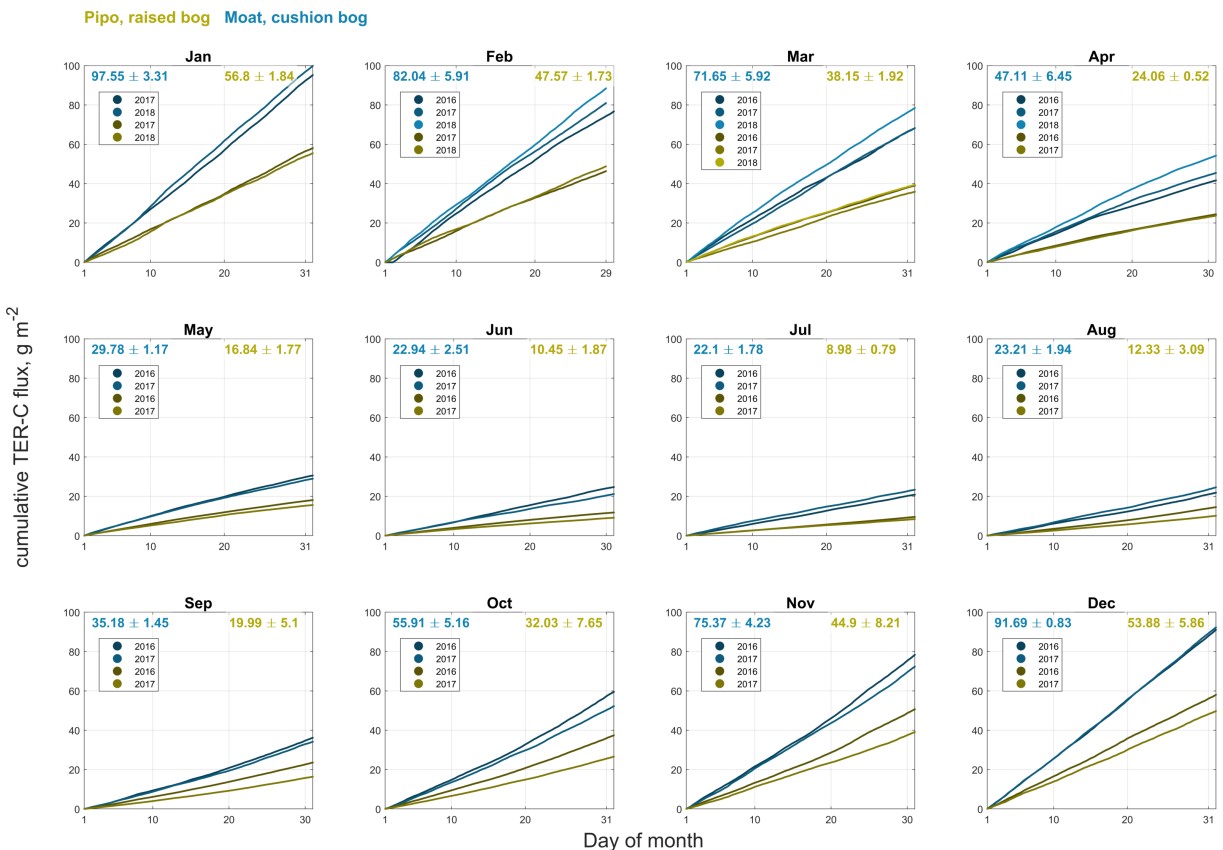

**Figure B2.** Cumulative total ecosystem respiration (TER) $CO_2$ fluxes including standard deviations for monthly intervals of all available full months within the years 2016, 2017 and 2018 at both investigation sites (yellow: Pipo, moss-dominated bog; blue: Moat, cushion plant-dominated bog).





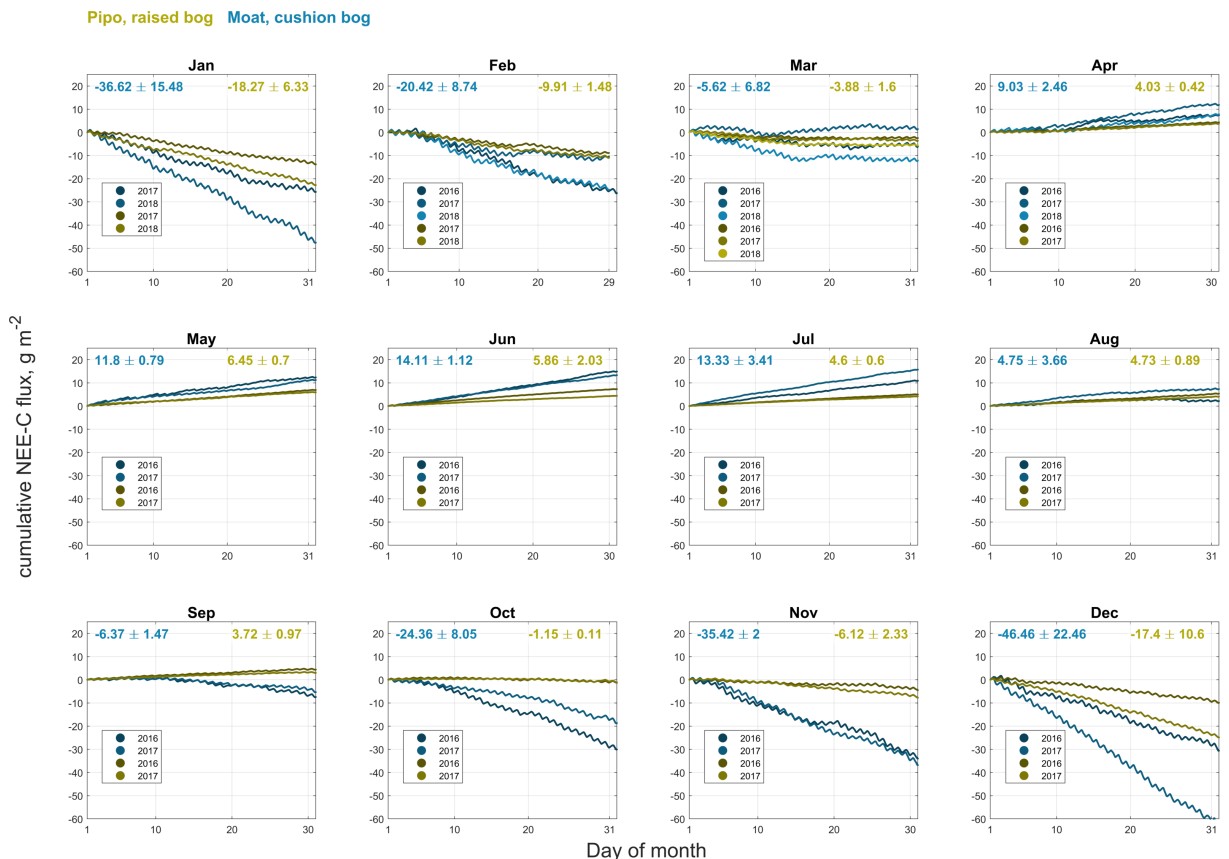

**Figure B3.** Cumulative net ecosystem exchange (NEE) $CO_2$ fluxes including standard deviations for monthly intervals of all available full months within the years 2016, 2017 and 2018 at both investigation sites (yellow: Pipo, moss-dominated bog; blue: Moat, cushion plant-dominated bog).



## Appendix C:  Cumulative temperature and radiation measures

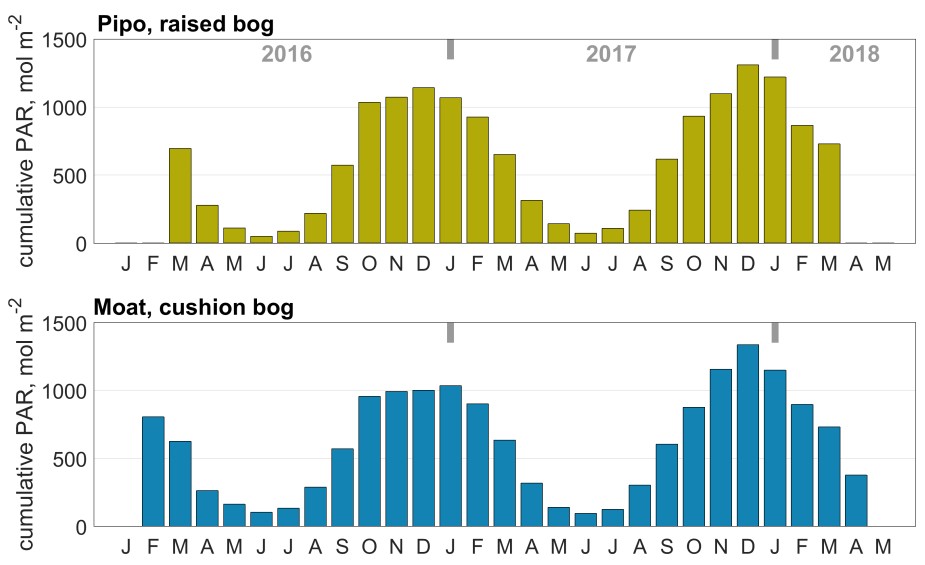

**Figure C1.** Monthly cumulated photosynthetically active radiation (PAR) at the two investigation sites

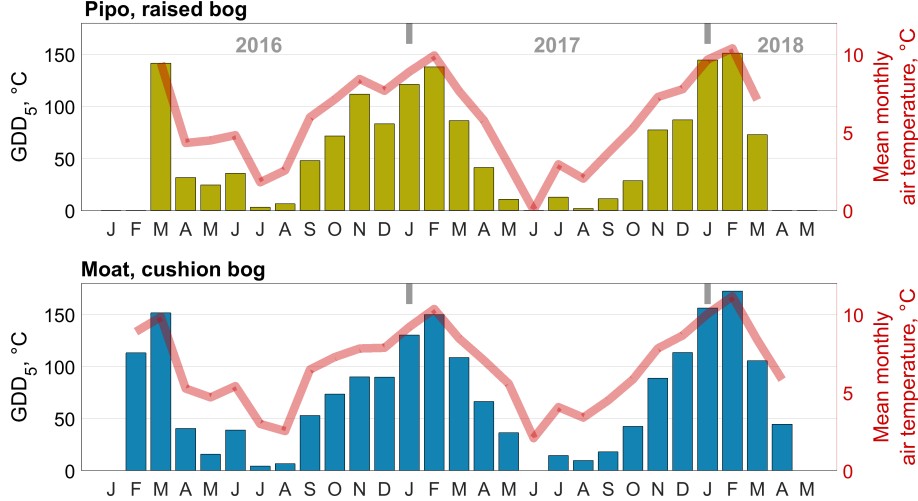

**Figure C2.** Monthly growing degree days (GDD) and mean air temperature at the two investigation sites.





## Appendix D: PAR$_{sat}$ calculation

$PAR_{\mathrm{sat}}$ denotes the radiation value at which the quantum use efficiency is reduced to one tenth of the initial quantum yield $\alpha$. We calculated the slope of $GPP(PAR)$ (see Eq. (1)) as

$$GPP'(PAR) = \frac{P_{max}^2 \alpha}{(P_{max} + \alpha PAR)^2} \tag{D1}$$

and used an attenuation factor $z_{sat}$ of 10 to scale $\alpha$ to an attenuated quantum use efficiency ($\frac{\alpha}{z_{sat}}$) that we set equal to $GPP'(PAR_{sat})$, yielding

$$\frac{\alpha}{z_{sat}} = \frac{P_{max}^2 \alpha}{(P_{max} + \alpha PAR_{sat})^2} \tag{D2}$$

which can be rewritten to:

$$PAR_{sat}^2 + \frac{2P_{max}}{\alpha} PAR_{sat} + P_{max}\left(\frac{1}{z_{sat}-1}\right) = 0 \tag{D3}$$

The positive solution of Eq. (D3) is given as Eq. (2) and was used to calculate $PAR_{\mathrm{sat}}$.

*Author contributions.* LK and VP conceptualized and administered the planning of the research activity and acquired the funds for it. AH, DH, LK, VP and SC conducted the field work. SC contributed data. AH and DH conducted literature research. DH analyzed the data, created visualizations and wrote the original draft. AH, DH, LK, VP and SC reviewed and edited the original draft.

*Competing interests.* No competing interests are present.

*Acknowledgements.* During field work, we were kindly hosted at and supported through staff and facilities of Centro Austral de Investigaciones Científicas (CADIC) operated by Consejo Nacional de Investigaciones Científicas y Técnicas (CONICET) in Ushuaia. We are especially grateful for the kind support of all members of the Laboratorio de Ecología Terrestre at CADIC. We very much appreciate being able to work in Parque Nacional Tierra del Fuego and want to thank APN (Administración de Parques Nacionales, Argentina). The Prefectura
Naval Argentina kindly granted permission to set up one of our wind turbines on their property, and their officers at the guard post in Moat



always welcomed us with most amicable hospitality. For supporting our fieldwork, we want to express our gratitude to Julio M. Escobar, Ing. Rodolfo Iturraspe, Ramiro Lopez, Prof. Dr. Christian Blodau, Prof. Dr. Till Kleinebecker, Wiebke Münchberger, Dr. Norman Rüggen, Tom Huber, Carla Bockermann, Isabella Närdemann, Juliane Kohlstruck, Laura Jansen, María Florencia Castagnani and María Noel Szudruk Pascual. We thank Gastón Kreps for sharing data from the meteorological long-term observations at CADIC. For his comprehensive

5 support during the technical planning of the measurement campaigns we thank Christian Wille. The project was supported through DFG (KU 1418/6-1).



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
