# Peer review of "Cushion bogs are stronger carbon dioxide net sinks than moss-dominated bogs as revealed by eddy covariance measurements on Tierra del Fuego, Argentina"

_Biogeosciences, 2019_

## Referee Comment (RC1) · Anonymous Referee #1 · 8 Jul 2019

The manuscript by Holl et al. (BG-2019-156) describes 2 years of $CO_2$ fluxes and their drivers in two different bog types in Argentina. The manuscript is written in detail, and data are thoroughly analyzed and shown with various angles. Flux data are scarce in this region, so the publication of this manuscript to Biogeosciences will be of a great interest to readers. However, I suggest some revisions for the publication.

The introduction (Section 1 and 2) contains a lot of information. It is informative to read the general characteristics of the study site, but some sentences are redundant and it can be written more concisely. In particular, Section 2.3 is interesting to read but it

does not add any information (i.e. it does not contain results of those past studies) to understand this study or the study site. In spite of the extensive introduction, some important components seem missing, for instance, why measuring multi-year $CO_2$ fluxes is important, why two bog types are chosen, what the definitions of raised/cushion bogs are, and etc. (with the given sentences in line#9 on page#2, cushion bogs seem to be defined as ombrotrophic bogs dominated by vascular plants). In addition, having this extensive introduction makes the readers expect it to continue throughout the results and discussion. However, it feels that some questions remain unanswered after reading the whole manuscript, such as 'why are the $CO_2$ flux patterns of these sites different from other bogs? Can it be from different vegetation communities, climate conditions, or the combination of both?' To improve this, the results should be discussed more thoroughly in relation to the topics raised in the introduction.

Section 4.4: 2 years of data is too small to discuss inter-annual variability. Will it be possible to find long-term climate data from those study sites and discuss what these 2-years of $CO_2$ data (response of $CO_2$ fluxes to climatic drivers) mean in relation to climatic variability? Also, Table 2 can be shown with actual numbers. It is easier to grasp the patterns with + and − signs, but less informative.

Section 4.7: What were the criteria for choosing these literatures for comparison (on page#21), similar plant types and climate conditions? Please add some more information to clarify this. On page#23, some more studies were mentioned, but the causes of the differences in $CO_2$ fluxes were not discussed comprehensively. Were the differences from the latitude only, or annual temperature, radiation, or something else? These questions continue to arise until the conclusion part, why these two bogs in Argentina show different $CO_2$ flux patterns than other bogs.

Usually either the names of the study site or plant types was used to refer to the sites, and having consistent names would be easier for the readers to follow. Especially, both the site names and plant types can be written in all the Tables and Figures.

---

## Referee Comment (RC2) · Nigel Roulet (Referee) · 9 Jul 2019

This is a very nice comparative analysis of the NEE, GPP, and ER of a southern hemisphere Sphagnum dominated bog and a cushion plant dominated bog. The authors clearly show that the cushion plant system has a greater NEE and they convincing show it is do to greater light use efficiency. They deduce this from eddy covariance measurements in the two systems. The authors provide the details in their methods and data processing – it all seem very sound.

[Figure]

The authors' data suggest there is something different in the photosynthetic efficiency of the cushion plants relative to Sphagnum. More correctly, they show the PAR saturation of Sphagnum occurs at a lower level of PAR. How Sphagnum photosynthesize is still a bit of a mystery. Are there any physiological and biochemical explanations why the cushion plants are adapted for higher light levels? Has any body done AiC curves for the cushion plants? These questions are at the root of the differences. Not suggesting the authors should know the answers but discussion along these lines would be useful.

The manuscript is very clean. Only editorial comments is Mer Bleue should have an 'e' at the end.

Nigel Roulet, McGill University, June 2019

---

## Author Comment (AC1) · 29 Jul 2019

Author reply to Referee comments from **Anonymous Referee # 1** from 8 July 2019 (https://doi.org/10.5194/bg-2019-156-RC1) on:

**Cushion bogs are stronger carbon dioxide net sinks than moss-dominated bogs as revealed by eddy covariance measurements on Tierra del Fuego, Argentina**
by David Holl et al.

Reviewer comments (RC)
Author comments (AC)
Mentioned line numbers refer to the originally submitted manuscript
Manuscript changes (MC)

The manuscript by Holl et al. (BG-2019-156) describes 2 years of CO2 fluxes and their drivers in two different bog types in Argentina. The manuscript is written in detail, and data are thoroughly analyzed and shown with various angles. Flux data are scarce in this region, so the publication of this manuscript to Biogeosciences will be of a great interest to readers. However, I suggest some revisions for the publication.

RC 1
The introduction (Section 1 and 2) contains a lot of information. It is informative to read the general characteristics of the study site, but some sentences are redundant and it can be written more concisely. In particular, Section 2.3 is interesting to read but it does not add any information (i.e. it d es not contain results of those past studies) to understand this study or the study site. In spite of the extensive introduction, some important components seem missing, for instance, why measuring multi-year CO2 fluxes is important, why two bog types are chosen, what the definitions of raised/cushion bogs are, and etc. (with the given sentences in line#9 on page#2, cushion bogs seem to be defined as ombrotrophic bogs dominated by vascular plants). In addition, having this extensive introduction makes the readers expect it to continue throughout the results and discussion. However, it feels that some questions remain unanswered after reading the whole manuscript, such as 'why are the CO2 flux patterns of these sites different from other bogs? Can it be from different vegetation communities, climate conditions, or the combination of both?' To improve this, the results should be discussed more thoroughly in relation to the topics raised in the introduction.
I extracted the individual points made in this paragraph and addressed them in detail below.

- In particular,  Section 2.3 is interesting to read but it does not add any information (i.e. it d es not contain results of those past studies) to understand this study or the study site.
  I agree that section 2 in general and section 2.3 in particular is a digression from the typical structure of a manuscript as most parts of it do not directly relate to the presented results. On the other hand, the fact that our study is the first comprehensive greenhouse gas flux investigation from Tierra del Fuego in our opinion justifies a brief review of the scientific exploration of the region to provide some context, which can be skipped by a reader who is mainly interested in our results. Anyway, if the editor considers the article to be too long for the journal, we would certainly accept the decision to remove this section. Results of other authors mentioned in section 2.3 which are relevant to our study are given later on in section 4.7.

- some important components seem missing, for instance,
  - why measuring multi-year CO2 fluxes is important, why two bog types are chosen,
    I added a paragraph detailing our reasons for the study design to the introduction (section 1):
    Our primary objective in this study is to describe the $CO_2$ flux dynamics of, in this respect, previously unstudied cushion bogs in the most general way feasible. To this end, we selected a second bog site for comparison, which represents a globally common type of ombrotrophic bog, and measured $CO_2$ exchange for more than two vegetation periods at both sites. This approach was thought to enable us to differentiate between variations of the $CO_2$ flux dynamics between cushion bogs and other global bogs that are mainly related to the varying climatic conditions on Tierra del Fuego and variations that are owed to the diverging traits of cushion plants in comparison to peat

mosses. Two years of data help to distinguish between generally valid properties and the effects of a potentially extreme year.

- what the definitions of raised/cushion bogs are, and etc. (with the given sentences in line#9 on page#2, cushion bogs seem to be defined as ombrotrophic bogs dominated by vascular plants). The introduction section on page 2 does not aspire to be complete with respect to bog type definitions. It highlights one important aspect of cushion bogs (that they are dominated by vascular plants) which sets them apart from most other global ombrotophic bogs. More complete descriptions and definitions of both studied bog types follow in section 2.2 (dedicated bog definition section) and 3.1 (site description). As Referee #1 mentions, sections 1 and 2 are already fairly extensive and contain some degree of redundancy. I therefore decided not to add more information to section 1.

- In addition, having this extensive introduction makes the readers expect it to continue throughout the results and discussion. However, it feels that some questions remain unanswered after reading the whole manuscript, such as 'why are the CO2 flux patterns of these sites different from other bogs? Can it be from different vegetation communities, climate conditions, or the combination of both?' To improve this, the results should be discussed more thoroughly in relation to the topics raised in the introduction.

Some of the issues raised in this comment have been addressed in response to RC3 by extending the introductory paragraph of section 4.7 (see below).

I added a more thorough discussion of possible reasons for NEE balance deviations between northern hemisphere moss-dominated bogs and the raised bog at Pipo from our study to the second paragraph of section 4.7

Average NEE component sums from Pipo are both of lower magnitude than reported by Moore et al. (2002) from Mer Bleue where |GPP| is about 25 % and TER about 30 % larger. Reasons for this lower productivity at the Fuegian raised bog in our study might be diverse. In terms of vegetation composition, a peculiarity of Fuegian moss-dominated bogs is the fact that only one *Sphagnum* species forms the different microforms (hummocks and lawns) that occur in these patterned bogs. *S. magellanicum* is, however, most likely not equally well adapted to all these topographic positions possibly diminishing the overall productivity of these bogs in comparison to more diverse northern mires. A further reason for the higher $CO_2$ uptake rates at Mer Bleue could be climatic distinctions. While the mean annual air temperature at Mer Bleue (6 °C) is similar to our 2017 value (5.3 °C) and to the long-term average for Ushuaia (5.5 °C), mean annual precipitation at Mer Bleue is with 943 mm nearly twice as high compared to what we measured between 01 March 2016 and 28 February 2017 (515 mm) at the Pipo raised bog. Additionally, evapotranspiration was nearly 200 mm higher during the same period, making our first observed year very dry and consequently imposing severe water stress on the vegetation. As already noted by Lehmann & Münchberger (2016) in 2015, we additionally observed a considerable, and in 2016 increasing, amount (between around 5 and 10 %) of surface being covered by lichens. Harris et al. (2018) showed that the presence of lichens indicate conditions that support near-zero C accumulation rates by limiting Sphagnum growth. Finally, our investigation of different photon use efficiency measures of the moss-dominated plant communities at Pipo indicate a non-optimal adaption of plant traits to the local conditions. While the vegetation at Pipo could use higher photon fluxes in winter than are actually present, the plant community is not able to efficiently use actually occurring high radiation input in summer (see Figure 6).

RC 2
Section 4.4: 2 years of data is too small to discuss inter-annual variability. Will it be possible to find long-term climate data from those study sites and discuss what these 2-years of CO2 data (response of CO2 fluxes to climatic drivers) mean in relation to climatic variability? Also, Table 2 can be shown with actual numbers. It is easier to grasp the patterns with + and − signs, but less informative.

I extracted the individual points made in this paragraph and addressed them in detail below.

- Will it be possible to find long-term climate data from those study sites and discuss what these 2-years of CO2 data (response of CO2 fluxes to climatic drivers) mean in relation to climatic variability?

Long-term climate data is not available for the exact locations of the study sites. For the raised bog at Pipo, long-term data does exist from relatively close locations (the city of Ushuaia and Ushuaia airport). It is however important to note that local weather conditions in the region can vary substantially due to the pronounced relief. For the cushion bog at Moat, no climate data apart from our records is available at all. This information is given in the site descriptions (page #6, line #30 onwards and page #8 line #1 onwards).

The first of the two observed years was comparably warm, in particular due to a very warm winter. This fact impacted both NEE components at the raised bog at Pipo. We did not miss out on discussing this topic in section 4.4 (page #16 line #18 onwards) and reiterate it in the conclusions (page #23 lines #25 onwards). With the available data, I honestly do not know what to add.

- Also, Table 2 can be shown with actual numbers. It is easier to grasp the patterns with + and – signs, but less informative.

The purpose of introducing Table 2 was to make the development of the different variables from the first to the second observed year easier to grasp. Taking the comment on Referee #1 into account, our intention seems to have come across. Because this way of presenting the results is less informative, we give actual numbers in Table 1. Together with the figures in Appendix C and the numbers given on page #16 from line #20 onwards we see no lack of information.

RC 3

Section 4.7: What were the criteria for choosing these literatures for comparison (on page#21), similar plant types and climate conditions? Please add some more information to clarify this. On page#23, some more studies were mentioned, but the causes of the differences in CO2 fluxes were not discussed comprehensively. Were the differences from the latitude only, or annual temperature, radiation, or something else? These questions continue to arise until the conclusion part, why these two bogs in Argentina show different CO2 flux patterns than other bogs.

I rewrote and extended the first paragraph of section 4.7. I added information about our reasons for choosing the particular literature for comparison and which properties the peatlands of the other studies share with and which distinguish them from the peatlands in this study.

We compared the NEE balances of this study to results from similar ecosystems. In case of the Pipo data set, we used long-term NEE observations from a northern hemisphere ombrotrophic raised bog in Canada and mean Holocene peat accumulation rates inferred from peat cores of other Patagonian bogs. We chose the Canadian Mer Bleue peatland as a reference because it represents a typical *Sphagnum*-dominated northern hemisphere bog and complete, multi-annual eddy covariance NEE, GPP and TER data have been published from this site. The average characteristics of northern hemisphere ombrotrophic bogs are well documented in these data sets and therefore enable a fairly robust estimation of differences with our $CO_2$ flux observations from a southern hemisphere *Sphagnum*-dominated raised bog. To evaluate if the results presented in this study can be seen as representative for Patagonian raised bogs in general or constitute an outlier because $CO_2$ flux dynamics are rather driven by local than by regional conditions, we used peat core data that has been acquired in similar bogs across southern Patagonia. For this comparison we took into account further components of the C balance as NEE can not directly be compared with peat accumulation rates. Comparisons with other global wetland types are less straight forward in case of our cushion bog data from Moat. Cushion bogs only exist on the southern hemisphere and have not been studied before in terms of their $CO_2$ flux dynamics. Similarities with other bog ecosystems from where NEE data has been published exist with respect to isolated features like the general geomorpholocial setting or the fact that they are located on the southern hemisphere and are vascular-plant dominated. For comparison, we used literature data from (I) Atlantic blanket bogs in Ireland that share their close proximity to the sea and the uneven subsurface relief they grow on with Patagonian cushion bogs and (II) NEE observations from Kopuatai bog in New Zealand which is also located on the southern hemisphere and is likewise dominated by vascular plants. Distinctions between these bogs and Patagonian cushion bogs are, however, pronounced. Even though Kopuatai bog in New Zealand is also vascular plant-dominated, the most frequent plant is a rush with a much higher growth height and distinctively lower root biomass than the cushion-forming plants *Astelia pumila* and *Donatia fascicularis*. Furthermore, Kopuatai bog is nearly 20° closer to the equator, decisively modulating the received radiation input in comparison to the cushion bog at Moat. Atlantic blanket bogs share the oceanic climate component and relief features of the underlying geology with the Fuegian cushion bog investigated in this study. Distinctions arise with respect to plant communities, as,

again, the occurring vascular plants do differ and a large belowground root biomass cannot be found in blanket bogs. Moreover, bryophyte cover is distinctly higher (Sottocornola et al., 2009) than in the cushion bog at Moat.

RC 4
Usually either the names of the study site or plant types was used to refer to the sites, and having consistent names would be easier for the readers to follow. Especially, both the site names and plant types can be written in all the Tables and Figures.
I updated Table 1 and Figures 5 and 6. Site names together with bog types are now used in all Figures and Tables consistently.

---

## Author Comment (AC2) · 29 Jul 2019

Author reply to Referee comments from **Nigel Roulet** from 8 July 2019
(https://doi.org/10.5194/bg-2019-156-RC2) on:

**Cushion bogs are stronger carbon dioxide net sinks than moss-dominated bogs as revealed by eddy covariance measurements on Tierra del Fuego, Argentina**
by David Holl et al.

Reviewer comments (RC)
Author comments (AC)
Mentioned line numbers refer to the originally submitted manuscript
Manuscript changes (MC)

This is a very nice comparative analysis of the NEE, GPP, and ER of a southern hemisphere Sphagnum dominated bog and a cushion plant dominated bog. The authors clearly show that the cushion plant system has a greater NEE and they convincing show it is do to greater light use efficiency. They deduce this from eddy covariance measurements in the two systems. The authors provide the details in their methods and data processing – it all seem very sound.

The authors' data suggest there is something different in the photosynthetic efficiency of the cushion plants relative to Sphagnum. More correctly, they show the PAR saturation of Sphagnum occurs at a lower level of PAR. How Sphagnum photosynthesize is still a bit of a mystery. Are there any physiological and biochemical explanations why the cushion plants are adapted for higher light levels? Has any body done AiC curves for the cushion plants? These questions are at the root of the differences. Not suggesting the authors should know the answers but discussion along these lines would be useful. The manuscript is very clean. Only editorial comments is Mer Bleue should have an 'e' at the end.

- I corrected the misspelling of Mer Bleue.
- To our knowledge, A/Ci data has so far not been published for *Astelia pumila* or *Donatia fascicularis.* Our group did actually attempt to estimate *in situ* leaf-scale photosynthesis of Astelia pumila using a Licor 6400 infrared gas analyzer, with which the determination of A/Ci curves would be possible in principle. However, fitting one of the rigid, relatively small and close to the ground growing leaves into the instruments measurement chamber while gaining signals consistently above the instrument noise level proved to be too challenging for the moment. We will need to adapt the methodology.
- I added a discussion about possible explanations for the effective light use of cushion plants to the end of the first paragraph of sectoin 4.6.

Reasons for the highly effective PAR use of *A. pumila* have been investigated by Fritz (2012) who found up to six times more leaf nitrogen per area compared to *S. magellanicum* caputila. Furthermore, Fritz (2012) found a high density of chloroplasts in cross sections of *A. pumila* leaves sampled at the Moat cushion bog on Tierra del Fuego. This notion is substantiated by our own (unpublished) data of chlorophyll content per gram dry weight, which was elevated by factors of up to ten in *A. pumila* leaves compared to *S. magellanicum* capitula. As a key competitive strategy of cushion plants for efficient nutrient recycling (Fritz, 2012) is the development of a dense and large root system, which contributes up to 90 % to their total biomass, the respiration cost that the plant imposes on itself by the maintenance of a large belowground part is high. The comparably high ecosystem respiration fluxes we determined in this study also point in this direction.